



# Mobile-Platform Measurement of Air Pollutant Concentrations in California: Performance Assessment, Statistical Methods for Evaluating Spatial Variations, and Spatial Representativeness

Paul A. Solomon[1], Dena Vallano[2], Melissa Lunden[3], Brian LaFranchi[3],
Charles L. Blanchard[4], Stephanie L. Shaw[5]

[1]Independent consultant, Henderson, Nevada, 89052, USA, formerly with U.S. Environmental Protection Agency, Office of Research and Development, Las Vegas, NV, 89119, USA
[2]U.S. Environmental Protection Agency, Region 9, Air and Radiation Division, 75 Hawthorne St, San Francisco, CA, 94105, USA
[3]Aclima, Inc., 10 Lombard St, Suite 200, San Francisco, CA, 94111, USA
[4]Envair, 526 Cornell Avenue, Albany, CA, 94706, USA
[5]Electric Power Research Institute, 3420 Hillview Ave, Palo Alto, CA, 94304, USA

*Correspondence to*: Charles L. Blanchard (charles.ld.blanchard@gmail.com)

**Abstract.** Mobile platform measurements provide new opportunities for characterizing spatial variations of air pollution within urban areas, identifying emission sources, and enhancing knowledge of atmospheric processes. The Aclima, Inc. mobile measurement and data acquisition platform was used to equip Google Street View cars with research-grade instruments. On-road measurements of air quality were made between May 2016 and September 2017 at high (i.e., 1-second [s]) temporal and spatial resolution at several California locations: Los Angeles, San Francisco, and the northern San Joaquin Valley (including non-urban roads and the cities of Tracy, Stockton, Manteca, Merced, Modesto, and Turlock). The results demonstrate that the approach is effective for quantifying spatial variations of air pollutant concentrations over measurement periods as short as two weeks. Measurement accuracy and precision are evaluated using results of weekly performance checks and periodic audits conducted through the sampler inlets, which show that research instruments in stationary vehicles are capable of reliably measuring nitric oxide (NO), nitrogen dioxide ($NO_2$), ozone ($O_3$), methane ($CH_4$) black carbon (BC), and particle number (PN) concentration with bias and precision ranging from <10 % for gases to <25 % for BC and PN at 1-s time resolution. The quality of the mobile measurements in the ambient environment is examined by comparisons with data from an adjacent (< 9 m) stationary regulatory air quality monitoring site and by paired collocated vehicle comparisons, both stationary and driving. The mobile measurements indicate that U.S. EPA classifications of two Los Angeles stationary regulatory monitors' scales of representation are appropriate. Paired time-synchronous mobile measurements are used to characterize the spatial scales of concentration variations when vehicles were separated by <1 to 10 kilometers (km). A data analysis approach is developed to characterize spatial variations while limiting the confounding influence of diurnal variability. The approach is illustrated using data from San Francisco, revealing 1-km scale enhancements in mean $NO_2$ and $O_3$ concentrations up to 117 % and 46 %, respectively, during a two-week sampling period. In San Francisco and Los Angeles, spatial variations up to factors of 6 to 8 occur at sampling scales of 100 – 300m, corresponding to 1-minute averages.



## 1 Introduction

In 2017, air pollution was responsible for nearly 5 million premature deaths worldwide, a 5.8 % increase from 2007 (Stanaway et al., 2018). Model projections indicate a possible doubling of premature mortality due to air pollution between 2010 and 2050 (Lelieveld et al., 2015). Multiple studies associate exposure to nitrogen dioxide ($NO_2$), particulate matter, carbon monoxide (CO), ozone ($O_3$), and sulfur dioxide ($SO_2$) with adverse health effects (Stieb et al., 2002; U.S. EPA, 2008; 2010a; 2010b; 2014; 2018; WHO, 2006).

Over the last 45 years, the public has relied on air quality information from stationary regulatory monitoring sites that are sparsely located throughout the U.S. With the advent of air quality monitoring equipment that can be placed across a range of locations using various sampling platforms (personal, stationary, and mobile), a greater spatial and temporal understanding of air quality can be obtained. With this information, members of the public can potentially reduce their health risks from air pollution. Improved understanding of spatial variations in air pollutant exposure is expected to yield increasingly accurate

estimates of the health effects of air pollution and is an important step in effectively reducing human exposure, acute and chronic health impacts, and premature mortality (e.g., Steinle et al., 2012). High spatial resolution measurements can reduce exposure misclassification and provide improved inputs for modeling. Spatially resolved air pollutant concentrations also aid in evaluating emission estimates and elucidating the effects of atmospheric processes on pollutant formation and accumulation. Urban air pollutant concentrations are known to vary by up to an order of magnitude over spatial scales ranging from meters

to hundreds of meters (Marshall et al., 2008; Olson et al., 2009; Boogaard et al., 2011). Previous efforts to characterize spatial variations in air pollutant concentrations have included near-roadway sampling (e.g., Baldauf et al., 2008; Karner et al., 2010), grid-based modeling (e.g., Marshall et al., 2008; Holmes et al., 2014; Friberg et al., 2016), land-use regression models (e.g., Gilbert et al., 2005; Henderson et al., 2007; Moore et al., 2007; Marshall et al., 2008; Hankey and Marshall, 2015), satellite data (e.g., Laughner et al., 2018), dense arrays of monitors (e.g., Blanchard et al., 1999; Kanaroglou et al., 2005; Kim et al.,

2018; Shusterman et al., 2018), and measurements made using mobile platforms (e.g., Brantley et al., 2014; Ranasinghe et al., 2016; Apte et al., 2017; Messier et al., 2018). The feasibility of deploying dense monitoring networks has increased with the availability of inexpensive sensors, although questions about sensor accuracy continue to be studied (e.g., Borrego et al., 2016; Castell et al., 2017; Li and Biswas, 2017; Schneider et al., 2017; Lim et al, 2019). Approaches that combine mobile monitoring with measurements made at stationary monitoring locations (Adams et al., 2012; Simon et al., 2018) or with modeling (Messier

et al., 2018) are being actively researched.

The Aclima, Inc. mobile measurement and data acquisition platform was previously used with Google Street View cars and equipped with research-grade instruments to measure air quality on city streets in Oakland, California between May 28, 2015 and May 14, 2016 (Apte et al., 2017) and through May 19, 2017 (Messier et al., 2018). The Oakland sampling campaign provided nearly complete coverage of all city streets with ~20 – 50 days sampling of each 30-meter (m) road segment, from

which high spatial resolution maps of average air pollution concentrations were constructed (Apte et al., 2017; Messier et al., 2018). The maps reveal persistent pollution patterns with small-scale variability attributable to local emission sources; 10 – 20



driving days reproduced spatial patterns with low bias and good precision (Apte et al., 2017). The Oakland results also demonstrate the efficiency of data-based mapping: using the data from all road segments obtained on only 4 – 8 drive days represented the full data set better than did measurements from a subset of road segments combined with a land use regression

– kriging model (Messier et al., 2018).

The Oakland study demonstrates an approach to mapping average air pollution concentrations within a defined geographical area by repeated sampling of each street. Mobile platform data from other locations are needed to better understand how wider coverage with more limited numbers of repeated samples within each neighborhood could be used in conjunction with data from stationary air quality monitoring locations to characterize neighborhood-scale variations. For example, new driving

strategies and analytical methods could help establish concentration decay rates with distance from roadways, comparability of pollutant concentrations among neighborhoods, and comparability of neighborhood concentrations to data from stationary regulatory monitors.

This study examines the field capabilities of mobile research-grade instruments used in varied settings. Future work will examine the capabilities of low-cost sensor data and will address the comparability of sensor and research-grade sampler data.

In this manuscript, instrument measurement accuracy and precision are evaluated using weekly performance checks, laboratory audits, and independent field audits conducted through sampler inlets. The quality of the mobile instrument measurements in the ambient environment is then examined by comparisons with adjacent (< 4 m) stationary air quality monitoring sites and by side-by-side paired vehicle comparisons. Mobile-platform measurements are compared to data from stationary air quality monitoring sites to evaluate and validate mobile-platform data and to ensure that the mobile platforms maintain high data

quality. The measurements obtained from replicate mobile platforms are compared using collocated vehicles that were operated while stationary and while driving; these results are used to establish the capabilities of the instruments for establishing high time-resolution spatial variations in pollutant concentrations. Finally, the mobile data are analyzed to examine the spatial representativeness of measurements made at stationary monitoring locations during selected time periods at a range of spatial scales (<1 km to >10 km).

The mobile measurements were made in various locations; an overview is available at https://blog.aclima.io/healthier-cities-through-data-ca-intro-6e9e22e00075 (last access, December 13, 2019). Because the driving routes were not designed to provide long-term repeated measurements for any of the locations, we did not focus on presenting pollutant maps. Rather, we examined measurement capabilities and developed statistical methods for analyzing the data. Data analysis methods were developed and applied to data subsets to exemplify approaches that are potentially applicable to larger data sets. Thus, some

results are illustrative rather than comprehensive. Since the measurements made during the study period were intended to address specific questions based on the results from specific sampling days, analyses are presented using different subsets of the data to address different questions. While performance evaluations and audit results are documented in this manuscript for all measured species, comparisons with stationary-monitor data, between-vehicle comparisons, and summaries of spatial variations are presented only for species that were measured using more than one platform (i.e., two vehicles or one vehicle

plus one stationary monitor).





## 2 Methods

### 2.1 Measurements

Measurements were made and processed by Aclima, Inc. All data are quality-assured by Aclima, Inc. at data quality levels 1 or 2 (qualified data level 1 [QD1] and qualified data level 2 [QD2]), as described in metadata documentation (Lunden and LaFranchi, 2017). The principal differences between QD1 and QD2 data are that the QD1 data include measurements made when the cars were parked overnight in garages and the QD2 data exclude calibration checks. Access to QD2 data is provided by Aclima, Inc. and Google, Inc. through the Google Cloud Platform using Google Cloud Shell and Google Big Query (Google, 2018). Aclima QD1 data were used for all analyses, because QD2 data (Google 2018) do not include the measurements made in the overnight parking garages; side-by-side comparisons of the measurements obtained when the cars were parked next to each other therefore required QD1 data sets (Aclima, 2018).

Street-level sampling was conducted in three California locations: San Francisco, Los Angeles, and smaller cities and nonurban areas within the northern San Joaquin Valley (Table 1). Measurements were made between ~ 9 a.m. and ~ 5 p.m. on weekdays, with additional sampling occurring in the San Francisco and Los Angeles parking garages before and after driving periods. Specific time periods were selected for analysis to represent data from different areas and to address individual research questions (Table 2). The selected periods do not represent the full set of driving routes in any of the areas but are instead intended to analyze routes that address the research objectives in Table 2, as discussed under results. Driving routes were mapped for visualization (supplement). For clarity, data are labelled by car names (Coltrane, Flora, Rhodes; these names do not duplicate the names of any stationary monitors).

During the Los Angeles sampling, the South Coast Air Quality Monitoring District (SCAQMD) conducted inlet audits and calibration checks when the sampling vehicles were parked adjacent to stationary air quality monitoring sites (Table 3). The SCAQMD also prepared 1-minute resolution data files for measurements made at various stationary air quality monitoring sites (Table 4; see also location map, Figure S1). Data from one of these dates and locations (LAXH, September 20, 2016) were suitable for collocated comparison with mobile measurements. The stationary-monitor data from W710 consisted only of 1-hour resolution $PM_{2.5}$ mass (Table 4), which was not measured by the mobile platforms, and no data were provided for the Santa Clarita site (Tables 3 and 4).

The Aclima mobile measurement and data integration platform consists of fast-response (<1 s to 8 s), research-grade analyzers providing data at 1-s (1-Hz) resolution. Details about the measurement techniques along with manufacturer specifications are provided in Lunden and LaFranchi (2017). The inlet and sampling manifolds were designed to minimize self-sampling as well as particle and gas phase sample losses. Separate inlet lines were used for particles (copper) and gases (Teflon™, a brand name of polytetrafluoroethylene). The gas-phase inlet line was set to a 90° angle to the direction of traffic and the particle and black carbon (BC) sampling inlet line faced forward. BC was measured using a photoacoustic extinctiometer, nitric oxide (NO) was measured using chemiluminescence, nitrogen dioxide ($NO_2$) was measured using cavity-attenuation phase-shift spectroscopy, ozone ($O_3$) was measured using ultraviolet (UV) absorption, and methane ($CH_4$) was measured using off-axis integrated cavity



output spectrometry. Particle number (PN) concentration was measured using an optical particle counter with particle counts

per liter (c L$^{-1}$) reported in 5 size ranges: 0.3 to 0.5 micrometer (μm) (PN$_{0.3-0.5}$), 0.5 to 0.7 μm (PN$_{0.5-0.7}$), 0.7 to 1.0 μm (PN$_{0.7-1.0}$), 1.0 to 1.5 μm (PN$_{1.0-1.5}$), and 1.5 to 2.5 μm (PN$_{1.5-2.5}$).

Each car recorded time using Network Time Protocol and times were reported to the nearest second universal time (UT). Timestamps were adjusted to account for residence time in the tubing and instrument response as described in Apte et al. (2017).

The gas-phase instruments received zero air and span gas weekly except for CH$_4$, which was checked weekly at a single concentration (2020 ppbv). Performance for the gas-phase measurements is expressed as bias and precision, defined according to the Data Quality Assessment guidelines used by the United States Environmental Protection Agency (EPA) (Camalier et al., 2007). For O$_3$, NO, and NO$_2$, the guideline analysis yields relative (in %) and absolute (in ppbv) contributions to uncertainties (Table 5). For CH$_4$, the analysis yields an absolute uncertainty for bias and precision of 66.7 ppbv (3.3 %), based

on reference measurements at 2020 ppb.

Additional uncertainties, which range from 1 % to 3.6 %, are associated with the accuracy of the calibration gas standards and the gas delivery/generation system. Field sampling uncertainties are discussed later.

The performance of the BC and PN instruments was evaluated from collocated parked vehicles (approximately weekly for PN and nightly for BC) since certified reference standards are not available for BC and PN. Both PN and BC instruments were

periodically returned to their respective manufacturers, typically once per year or when the results of ambient collocations indicated substantial drift of one car relative to the other(s) or other diagnostic checks indicated that service was required. Table 6 shows the results of evaluations performed between May 2016 and August 2017.

We calculate BC LOD using data reported while the instrument is performing an internal zero, which occurs every 10 minutes for 60 seconds. This value is typically in the range of 0.2-0.3 μg m$^{-3}$ for the 1-Hz data while the cars are parked. For vehicles

in motion, we estimate 1-Hz LOD values of 0.4 μg m$^{-3}$ for vehicle speeds less than 5 m s$^{-1}$ and 0.8 μg m$^{-3}$ for vehicle speeds greater than 5 m s$^{-1}$.

## 2.2 Location Uncertainty

Location uncertainty was determined as the variability of recorded positions when vehicles were parked. The vehicles did not have designated spaces to which they always returned. Therefore, variances and standard deviations of parked-vehicle east-

west and north-south GPS locations were determined by vehicle, date, and time of day (i.e., before and after each daily drive). Composite east-west and north-south standard deviations were then determined from individual variances weighted by sample numbers. Composite variances were converted to location uncertainty (twice the square root of the sum of the east-west and north-south composite variances). The observed 2σ location uncertainty for vehicles parked in the San Francisco parking structure was ± 6.0 m, comparable to the GPS manufacturer specifications (5 m). The location uncertainties for vehicles parked

in the Los Angeles parking structure were larger (± 12.2 m at 1 s resolution and ± 11.5 m for 1-minute averages). The GPS location uncertainties therefore impose inherent limits to the spatial resolution of the data on the order of 10 m.



### 2.3 Comparisons between Measurement Platforms

For ambient comparisons between vehicles or between vehicles and stationary monitors, our approach for computing comparability necessarily differs from EPA guidelines for determining precision and bias, because neither vehicle nor stationary monitor measurements are certified target concentrations. Thus, comparability must be determined in terms of the differences between measurements made by different vehicles or between vehicle and stationary-site data, which yields instrument-to-instrument comparability. Data files were merged by time of day using either 1-s and 1-minute resolution measurements. The merged data were used to determine time-matched paired differences and to evaluate intervehicle measurement variabilities as functions of ambient concentration, intervehicle distance, and vehicle speed. Paired differences were evaluated for bias of one measurement relative to another. The variabilities of the paired differences relative to the means of the paired differences were also calculated. The computational approach was necessarily limited to parameters that were measured on each of two platforms (e.g., two cars or one car plus one stationary monitor). BC and $CH_4$ were each measured by only one vehicle while operating (during drives, one vehicle was equipped with a BC sampler and the other with a $CH_4$ instrument). Therefore, it was not possible to compare BC or $CH_4$ concentrations between operational vehicles (as previously noted, however, BC and $CH_4$ instruments were installed on multiple vehicles and used to establish parked-vehicle instrument-to-instrument bias and precision). BC and $CH_4$ data were not available from stationary monitors.

### 2.4 Statistical Metrics

Various statistical metrics were computed to evaluate the comparability of time-paired measurements between vehicles or between vehicles and stationary monitors. These metrics include mean differences and fractional (relative) mean differences:

Mean Difference (MD) = $\mu_{A-B}$ = mean$(X_A - X_B)_i$ = Mean (Car A – Car B Difference)         (1)

        where $\sigma_{A-B}$ = standard error (SE) of the mean of $(X_A - X_B)_i$ ,

        "i" denotes the "i$^{th}$" measurement of n paired measurements,

        SE = $(\sqrt{n})^{-1} \times$ standard deviation of $(X_A - X_B)$

Fractional (relative) Mean Difference (FMD) = $\mu_{A-B} / \mu_{AB}$         (2)

        = Mean (Car A – Car B Difference)/Mean of Car A and Car B Mean Concentrations

        $\sigma^2_{FMD} = \{(\sigma_{A-B} / \mu_{AB})^2 + (\sigma_{AB} \times \mu_{A-B} / \mu_{AB}^2)^2\} = FMD^2 \times \{(\sigma_{A-B} / \mu_{A-B})^2 + (\sigma_{AB} / \mu_{AB})^2\}$

        where $\mu_{A-B}$ = mean$(X_A - X_B)_i$ and $\sigma_{A-B}$ = standard error (SE) of the mean $(X_A - X_B)_i$,

        $\mu_{AB} = \{(1/2) \times (\mu_A + \mu_B)\}$ and $\sigma^2_{AB} = \{(1/4) \times (\sigma^2_A + \sigma^2_B)\}$

Fractional Absolute Mean Difference (FAMD) = $| \mu_{A-B} | / \mu_{AB}$         (3)

        = |Mean (Car A – Car B Difference)|/Mean of Car A and Car B Mean Concentrations

        $\sigma^2_{FAMD} = FAMD^2 \times \{(\sigma_{A-B} / \mu_{A-B})^2 + (\sigma_{AB} / \mu_{AB})^2\}$

        where $\mu_{A-B}$ = mean$(X_A - X_B)_i$ and $\sigma_{A-B}$ = SE$(X_A - X_B)_i$ and

        $\mu_{AB} = \{(1/2) \times (\mu_A + \mu_B)\}$ and $\sigma^2_{AB} = \{(1/4) \times (\sigma^2_A + \sigma^2_B)\}$





Fractional Mean Absolute Difference = FMAD = $\mu_{|A\text{-}B|}$ / $\mu_{AB}$ (4)

200          = Mean |Car A – Car B Difference|/Mean of Car A and Car B Mean Concentrations

$\sigma^2_{FMAD}$ = FMAD$^2$ × {($\sigma_{|A\text{-}B|}$/ $\mu_{|A\text{-}B|}$ )$^2$ + ($\sigma_{AB}$ / $\mu_{AB}$ )$^2$}

where $\mu_{A\text{-}B}$ = mean|$X_A - X_B$ |$_i$ and $\sigma_{A\text{-}B}$ = SE|$X_A - X_B$ |$_i$ and

$\mu_{AB}$ = {(1/2) × ( $\mu_A$ + $\mu_B$ )} and $\sigma^2_{AB}$ = {(1/4) × ($\sigma^2_A$ + $\sigma^2_B$ )}

The variances $\sigma^2_{FMD}$, $\sigma^2_{FAMD}$, and $\sigma^2_{FMAD}$ are derived from standard statistical formulae (e.g., variance of (X × Y) = {($\sigma_X$ × $\sigma_Y$)$^2$

+ ($\sigma_X$ × $\mu_Y$)$^2$ + ($\sigma_Y$ × $\mu_X$)$^2$}, http://www.odelama.com/data-analysis/Commonly-Used-Math-Formulas/, last access September

27, 2019) by transforming variables (X/Z = X × Y, Y = Z$^{-1}$) and by making two assumptions: (1) the numerator and

denominator (e.g., $\mu_{A\text{-}B}$ and $\mu_{AB}$) are independent (implying zero covariance between differences and means), and (2) higher-

order terms ($\sigma^2_X$ × $\sigma^2_Y$ ) are small compared with ($\sigma^2_X$ × $\mu^2_Y$) and ($\sigma^2_Y$ × $\mu^2_X$) (because the standard errors [$\sigma_{A\text{-}B}$ and $\sigma_{AB}$] are

based on large sample sizes, e.g., n > 1000, and standard errors are inversely proportional to the square root of sample size).

Standard errors are the appropriate measure of the variability of mean concentrations and differences, such as those defined

here, whereas standard deviations are appropriately used to quantify the variability of individual measurements (Results and

Discussion).

The preceding equations, while expressed as car-to-car comparisons, are readily applied to other comparisons, e.g., vehicle-

to-stationary monitor. If one measurement (e.g., measurement A) is defined as a reference standard, then the term $\mu_{AB}$ in the

denominator of the expressions for FMD, FAMD, and FMAD may be appropriately replaced by the reference mean ($\mu_A$). Mean

differences (MD) are used when absolute comparisons (i.e., retaining concentration units) are informative. Fractional

differences are useful for establishing vehicle-to-vehicle or vehicle-to-monitor differences relative to the magnitudes of the

mean concentrations.

The FMD retains sign, i.e., indicates if $\mu_A$ > $\mu_B$. This metric is useful when the sign is important for identifying which

instrument (e.g., mobile or stationary) or which location records higher concentrations. The FAMD and FMAD are useful if

the sign of the difference is not meaningful. The sign is usually not relevant, for example, in the analysis of intervehicle

measurement differences as a function of the distance between the vehicles (see results and discussion), in which the objective

is to characterize the rate at which measurement comparability decays with distance. The FAMD is simply the absolute value

of the FMD and both metrics approach zero when individual paired measurement differences tend to average out over a set of

samples. In contrast, the FMAD provides a measure of the variability of individual measurements because it averages absolute

values of concentrations. The FMAD is relevant to understanding the comparability of high-resolution (e.g., 1 s)

measurements, whereas the FAMD is a measure of the comparability of a time- or space-average determined from individual

measurements.

Performance audits (Tables 5 and 6) indicate that fractional differences (FAMD) exceeding ~0.1 (10 %) for gases and ~0.2

(20 %) for PN are, in general, likely to be physically meaningful relative to measurement uncertainties (bias and precision are

each < 5 % for gases at concentrations > 2 – 24 ppbv; 7 – 26 % for PN and BC). Only the two largest PN size ranges exhibit

bias exceeding 20 % (Table 6). Combining bias and precision indicates a total uncertainty of ~10 % for gases and ~20 % for



$PN_{0.3-0.5}$. In operation, the comparability of measurements made in moving vehicles differs from those made in parked collocated vehicles (see results and discussion), so we utilize a higher threshold (i.e., 20 %) for establishing true spatial

variations even for gas-phase species.

## 3 Results and Discussion

Mean concentrations during example study periods are summarized in Table 7. Subsequent analyses in this section, which depend on the availability of measurements from two or more sampling platforms, focus on NO, $NO_2$, $O_3$, and $PN_{0.3-0.5}$. These pollutants are of interest because they are measured with differing accuracies, they exhibit differing degrees of spatial variation,

and they vary in their degree of atmospheric chemical processing. NO is a primary pollutant and $NO_2$ forms rapidly (i.e., minutes) from NO. $NO_2$ formation and $O_3$ loss are linked through the rapid reaction of NO with $O_3$ to form $NO_2$; Seinfeld and Pandis (2016) calculate a 1/e lifetime for NO of 42 seconds at 50 ppb $O_3$. $O_3$ formation and accumulation occurs more slowly (i.e., hours) from $NO_2$ and volatile organic compounds (VOCs) in the presence of ultraviolet (UV) radiation (Seinfeld and Pandis, 2016). $PN_{0.3-0.5}$ is the smallest size fraction that was measured, present in the highest numbers (83 % of PN, Table 7),

and is likely indicative of newly aged particles from fresh motor-vehicle emissions (Zhang and Wexler, 2004; Zhang et al., 2004; Zhu et al., 2002).

The fraction of PN in the 0.3 – 0.5 μm size fraction was lower in spring (60 % in San Francisco, May 2017 and 72 % in the San Joaquin Valley, March 2017) and higher in summer (90 % in Los Angeles, August 2016) and autumn (86 % in Los Angeles, September 2016 and 84 % in the San Joaquin Valley, November 2016) (Table 7). Although differences in the PN

distributions possibly reflect spatial variability, they more likely reflect seasonal variations in PM composition: the observed variations in PN distributions are consistent with past studies that indicate the importance of PM nitrate ($NO_3$) found in larger (> 0.5 μm) size fractions primarily as ammonium nitrate in California during cooler months (e.g., Herner et al., 2005).

Mean concentrations of gases were comparable among the study locations and periods (Table 7). $O_3$ concentrations were highest in Los Angeles in August near downtown (south of the CELA site, Figures S6 and S7) followed by concentrations in

September in west Los Angeles near the WSLA site (Figure S8) and near Los Angeles airport (near the LAXH site, Figure S3). Mean $O_3$ in the remaining locations (SJV and SF) fall within a narrow range (23 – 29 ppbv) and are only a factor of less than 2 lower than in Los Angeles. Mean concentrations of $NO_2$ also vary by a factor of 2 with highest concentrations near the LA airport and lowest concentrations in SF (Table 7). Concentrations of NO are highest by a factor of about 2 in Los Angeles near the airport and in the SJV in November during mostly freeway driving. At all locations studied, typical NO-$NO_2$-$O_3$

chemistry was observed with higher NO and $NO_2$ concentrations and lower $O_3$ levels near mobile emission sources. Mean methane concentrations were low (~ 2 ppmv) during all periods and varied among areas within <0.1 ppmv. As with PN, these average concentrations likely vary due to time of year, location relative to source emissions, and chemical processing.



### 3.1 How Well Do Measurements in the Mobile Platforms Compare to the Inlet Audits?

Calibration checks (zero and span) were conducted in the field using SCAQMD equipment and standards; these checks were
compared with Aclima calibration checks that were made before, during, and after the period when vehicles drove in the Los
Angeles (Table 8). The SCAQMD and Aclima checks were comparable and indicate that measurements of the tested gas-phase
species (NO, $NO_2$, and $O_3$) maintained accuracy and replicability in the field during the Los Angeles driving routes. The Los
Angeles drives followed the same field protocols as the drives in San Francisco and the San Joaquin Valley. The cross-lab
differences between the Aclima and SCAQMD calibration checks (defined as the lab-to-lab differences in the mean relative
differences from target concentrations averaged over all calibration checks) were -5 % ± 2.0 % for NO, -1.5 % ± 1.0 % for
$NO_2$, and +0.5 % ± 1.3 % for $O_3$ (not tabled).

### 3.2 How Similar are Concentrations Obtained from Collocated Vehicles when Parked and When Moving?

Car-to-car comparisons were made to evaluate the comparability of collocated ambient measurements made while the vehicles
were parked and while driving (Table 9). These measurement differences reflect a combination of instrument and ambient
sampling uncertainties; for moving vehicles, differences may also reflect spatial variability, depending on measurement
integration times relative to intervehicle distances. The comparisons are expressed as mean car-to-car differences plus-or-
minus 1 standard deviation of the paired 1-s differences, yielding metrics for car-to-car measurement bias and variability,
respectively, averaged over ~1000 – 50,000 paired differences.

The observed mean paired differences between parked vehicle measurements were 0.2 – 3.9 ppbv for NO, 0.3 – 1.9 ppbv for
$NO_2$, and 0.8 – 4.5 ppbv for $O_3$ (Table 9). The corresponding FAMD (absolute values of mean differences divided by mean
concentrations) range from 0.03 – 0.24 (3 % to 24 %) for gases and 0.04 – 0.22 (4 % to 22 %) for PN. These differences are
comparable to, or larger than, instrumental bias and precision (<5 % each for gases at concentrations > 2 – 6 ppbv, Table 5;
10 – 11 % for $PN_{0.3 - 0.5}$, Table 6). For gases and PN, the variabilities (standard deviations) of the 1-s paired differences exceed
the mean differences ( except $O_3$ during the SJV sampling period of November 16 – 23, 2016), which is expected because
instrumental variations average toward zero when instruments are unbiased with respect to each other. The mean paired
differences varied among individual sampling days (Figure S2). Between-vehicle 1-s variability is higher in closely-spaced
moving vehicles than in stationary vehicles, especially for $NO_2$ (Table 9; note that this comparison could not be made for NO).
We interpret this difference as indicating that moving vehicles sampled heterogenous parcels of air and the intervehicle
measurement differences are thus due to fine-scale spatial variability.

### 3.3 How Similar are Mobile Reference Concentrations to Stationary Monitor Data?


On September 20, 2016, two sampling cars parked next to the monitor at LAXH (Tables 3 and 4, Figures S3 and S4). Relative
to the ground-level position of the stationary monitor probe (located inside a fenced enclosure), the vehicles alternated positions
from closer when audited (Coltrane 6.6 m from LAXH, Flora 8.5 m from LAXH) to further when sampling (Coltrane, 24.1 m



for 1 hour; Flora, 18.5 m for 2 hours) as determined from GPS coordinates for the monitor and vehicles. The heights of the
LAXH instrument probes are 4.2 m above ground level (SCAQMD, 2018a), whereas the mobile sampler inlet heights are 2 m
above ground level. The monitoring instruments at LAXH are in a vacant field north of Los Angeles International Airport
(Figure S4). The site is surrounded by several schools to the NE, N, and NW with residential communities (Playa Del Rey and
Westchester) north of the airport and further away surrounding the site. The closest communities include homes and 2 – 4
story apartments. Minimal traffic is expected immediately adjacent to the site.

The mobile platforms recorded mean concentrations of NO, $NO_2$, $O_3$, and $O_x$ (= $NO_2$ + $O_3$) that were comparable to LAXH
monitor concentrations:  most mean paired differences between mobile-platform and LAXH concentrations were less than 10
% of the average concentrations (Table 10). Time series of 1-minute Flora, Coltrane, and LAXH measurements show
agreement (Figure S5) (mean Flora – Coltrane distances were 12.2 and 20.2 m). $CH_4$ concentrations can be a potential tracer
of fresh motor-vehicle emissions and all NO values correlated with Coltrane $CH_4$ concentrations ($r^2$ = 0.84 to 0.87; Flora did
not report $CH_4$).

### 3.4 How Large are the Differences between Mobile Reference Concentrations and Stationary Monitor Data When the Cars are Not Close to Monitors?

Spatial variation is defined by differences in time-synchronous measurements made in differing areas. To interpret the paired
differences as spatial variation, rather than measurement uncertainty, we refer to the preceding analyses of instrument and
sampling performance in audit tests (Tables 5 and 6) and collocated vehicles (Table 9). As previously noted, the results for
measurement bias and precision (Tables 5 and 6) and for comparability of collocated vehicles (Table 9) lead us to define
FAMD > 0.2 (20 %) as an indicator that spatial variations exceed measurement and sampling uncertainties. The intent of the
analyses in this section is to help elucidate the spatial scales over which stationary-monitor and mobile-platform data represent
ambient concentrations.

**3.4.1 Los Angeles, August 2016**

Between August 3 (the first complete Los Angeles driving day) and August 12, the two vehicles traversed different
neighborhoods south of the central Los Angeles stationary monitor (CELA, Table 4) at varying speeds between 9 a.m. and 6
p.m. at car-monitor distances ranging from 1 to 7 km (Figure 1). The monitoring instruments at CELA are located on a rooftop
of a two-story building and the heights of various instrument probes range from 11 to 12 m above ground level (SCAQMD,
2018b). Driving routes for the first sampling day (August 3) are shown in Figure S6; most of the routes on other dates were
similar. In general terms, the US-101 and one section of the I-5 freeways run across the southern border of the sampling area;
the area sampled is split by a N-S portion of I-5 and bordered on the north by I-10. The I-10 freeway is situated between CELA
and the measurement area. For comparison with the 1-minute resolution CELA data, 1-minute average concentrations were
created from the 1-s mobile-platform data. Because driving speeds averaged ~ 2 – 5 m s$^{-1}$ (Figure 1), the typical distances
traveled in one minute were ~100 – 300 m. The 1-minute average positions of the mobile sampling are visibly discrete (Figure





S6). While in motion, generally beginning after 9:00 a.m. and ending between 5:00 and 6:00 p.m., the cars recorded higher concentrations of NO and $NO_2$ than the CELA stationary air monitor did, likely due to the proximity of fresh vehicle emissions experienced by street-level sampling in the vehicles (Figure 1). During the driving hours, the vehicles recorded lower levels of $O_3$ than CELA did (Figure 1). As noted in the previous comparison of collocated and stationary-monitor data, much of this

difference is attributable to street-level reaction of fresh NO emissions with $O_3$; this interpretation is supported by the closer agreement between cars and CELA of $O_x$ than $O_3$ (Figure 1).

Between-vehicle paired comparisons were determined as differences between time-synchronous 1-min mobile concentrations for August 3 – 12 (near CELA), which were then averaged over 0.5 km bins (0 – 0.25 km, 0.25 – 0.75 km, etc.) (Figure 2). The bin-average FAMDs ranged from 0.02 (2 %) at 0.125 km to 0.14 – 0.44 (14 – 44 %) at 4.5 – 5.5 km (mean = 0.12, or 12

%, over all bins) for $NO_2$ and from 0.006 (0.6 %) at 0.125 km to 0 – 0.07 (0 – 7 %) at 4.5 – 5.5 km (mean = 0.02, or 2 %, over all bins) for $O_3$. For these two pollutants, the mean differences among streets and neighborhoods were therefore small (12 % and 2 %, respectively, at 0.125 – 5.5 km spatial scale). For NO, bin-average FAMDs were larger and ranged from 25 % at 0.125 km to 4 – 75 % at 4.5 – 5.5 km.

The intervehicle differences averaged over distance bins concisely summarize large numbers of measurements but this

averaging could mask finer spatial variations of possible interest. The results obtained for bin averages were examined for higher variability on smaller spatial scales. We compared the standard deviations of the mean intervehicle concentration differences to the corresponding mean concentrations to characterize variability within the spatial averages. These ratios (standard deviation of intervehicle difference/mean concentration) ranged from 0.4 to 1.0 (average = 0.5) for $NO_2$. Within the binned intervehicle averages, therefore, vehicle-to-vehicle $NO_2$ concentration differences varied by up to a factor of two (twice

the standard deviation of the mean differences) times the mean observed concentrations. For NO, the ratios ranged from 1.2 to 4.0 (average = 2.8), indicating that vehicle-to-vehicle NO concentration differences varied by up to a factor of six (two standard deviations) within the binned intervehicle averages.

The number of particles in the size range 0.3 to 0.5 μm exhibited FAMDs exceeding 0.2 (20 %) that were less variable than the NO FAMD. Both NO concentrations and particle numbers likely varied as the vehicles sampled different streets and

neighborhoods and experienced differing levels of fresh emissions at any given time (e.g., Figures S6 and S7). The peak in the NO FAMD at 3 and 3.5 km corresponds to mean NO concentrations of 6.6 and 8.1 ppbv, respectively for Flora and mean NO concentrations of 14.8 and 15.3 ppbv, respectively, for Coltrane. Many of the 85 and 120 1-minute differences in these two bin averages correspond to cases where Coltrane sampled close to the confluence of the Santa Anna and Golden State freeways while Flora collected data further from freeways (Figure S7). An approach to identifying high-concentration locations is

illustrated later in the discussion of data from San Francisco.

The NO FAMD for car-CELA comparisons largely exceeded 1; the $NO_2$ and $O_3$ FAMDs were less than 0.5 and 0.2, respectively, at most car-CELA distances (Figure 3). Although the two cars drove different routes, the two car-CELA comparisons were similar (Figure 3). The representativeness of CELA and other sites is discussed below.





### 3.4.2 Los Angeles, September 2016

Driving routes were near (0.5 to 5 km) the west Los Angeles stationary monitor (WSLA, Table 4) on four of the 14 days between September 12 and 30 (Figure S8 for September 13 and 19; similar routes were driven on September 26 and 29). Drives began at ~9 a.m. and ended by 5 p.m. LDT. Because only one car drove near WSLA on each of the four days, only car-to-WSLA comparisons are presented. The monitoring instruments at WSLA are located on the roof of a trailer on the grounds of the VA hospital and the heights of the instrument probes are 4.2 m above ground level (SCAQMD, 2018c) (Figure S9). The

monitor is located <600 m west of I-405 and about 200 m south of a major arterial, Wilshire Blvd. The immediate surrounding area to the north and south is grass with some trees, and slightly further out the area is primarily residential multistory (2 – 3 stories) apartment buildings.

The mobile platforms recorded substantially (between 70 % up to a factor of 32) higher concentrations of both NO and $NO_2$ than WSLA while the cars drove from the parking garage on the Santa Monica freeway to the neighborhood destinations

(Figure 4, WSLA-car distances > 5 km). Even at distances < 0.5 km up to 5 km from WSLA, the mobile platforms recorded higher concentrations of NO and $NO_2$. However, mean car and WSLA $O_x$ concentrations at distances < 10 km were more similar than were corresponding car and WSLA concentrations of $NO_2$ and $O_3$ (Figure 4). For NO and $NO_2$, the FAMD exceeded 1.5 and 0.4, respectively, at all distances outside the parking garage (Figure 5). During part of their routes, the cars sampled adjacent to the San Diego (I-405) freeway, which likely contributed to higher mean NO and $NO_2$ concentrations for

the mobile platforms. The WSLA monitoring site (grounds of VA hospital) has a middle scale zone of representation (100 m to 0.5 km) for $NO_2$ (Table 4), consistent with our results. For $O_3$ and $O_x$, the FAMD were < 0.2 and < 0.05, respectively, within 5 km of WSLA.

### 3.5 How Large are the Differences between Pollutant Concentrations Reported by Vehicles Operating in Different Neighborhoods?

Answers to this question help identify neighborhoods where pollutant concentrations are typically higher than occur elsewhere. In such neighborhoods, air pollutant exposures are potentially higher than levels measured by regulatory monitors. Analyses are useful for identifying areas experiencing higher pollutant concentrations and, potentially, locating long-term monitors for characterizing higher pollution impacts.

### 3.5.1 San Francisco, May 2017

Measurements made by paired vehicles operating in different neighborhoods of San Francisco between May 1 and 12, 2017, are used to illustrate short-term (two week) neighborhood-scale spatial variability. Example driving routes are shown as 1-s averages for one day in Figure S10. The 1-s data were aggregated to 1-minute averages and the one-minute averages for all routes for May 1 - 12 are depicted in Figure 6a. Different routes were taken on different days to obtain measurements in different neighborhoods in San Francisco. Since the averaging driving speeds between May 1 and 12 were 4.5 and 4.8 m s⁻¹



for Coltrane and Flora, respectively, the positions shown in Figure 6a represent the midpoints of segments averaging 270 – 290 m.

One-minute averages were next averaged spatially to the nearest kilometer separately for each car (Figure 6b), which is a spatial scale corresponding to about a 3-minute average. However, the sampling times of the 1-km average concentrations varied by up to six hours among locations, which confounds spatial with diurnal variability. Instead of analyzing 1-km average

concentrations by vehicle, therefore, each 1-minute average was paired with the corresponding 1-minute average reported by the other vehicle and synchronous concentration differences were determined. When these synchronous differences are averaged to 1-km resolution, they represent the average enhancement or deficit of a pollutant at a given 1 km location when compared to simultaneous measurements made elsewhere, i.e., the average excess or deficit relative to co-measured concentrations (Figure 6c and 6d). This approach permits consideration of spatial variations in a manner that limits the

confounding influence of diurnal variability and provides a better relative comparison of pollutant levels among neighborhoods.

One-km averages consisting of fewer than ten 1-minute data points were excluded, yielding 97 of 236 possible spatial averages for $NO_2$ and 107 of 271 possible spatial averages for $O_3$. The decision to exclude 1-km averages consisting of fewer than ten 1-minute data points was based on the high standard errors of such averages (e.g., > 0.2 for the $NO_2$ FAMD when n < 10). The

number of 1-minute averages within each 1-km average ranged from 10 to 95 (i.e., 60 – 5700 1-s averages); for, the 1-km average covering the parking garage, there were 1813 and 2520 1-minute $O_3$ and $NO_2$ averages, respectively.

For both $NO_2$ and $O_3$, most 1-km average concentration differences exceeding 2 ppbv (or < - 2 ppbv) were statistically significant (i.e., the interval of the mean difference ± 2 standard errors of the mean did not cover zero); most differences in the range between -2 and 2 ppbv were not statistically different from zero (Figure 6c and 6d). The figures exclude the few larger

differences that were not statistically different from zero (7 $O_3$ and 4 $NO_2$ averages). Both fractional differences and the signs (excess or deficiency) of the differences are of interest; therefore, the mean fractional differences are expressed as FMD rather than FAMD (Figures 6e and 6f) since the sign of the difference is important. For $NO_2$, FMD exceeding 0.5 (or < -0.5) were statistically different from zero; for $O_3$, FMD exceeding 0.05 (or < -0.05) were statistically different from zero. The contrast in the detectability of statistically significant fractional $NO_2$ and $O_3$ differences between vehicles (FMD) is pronounced but

readily explained: the average intervehicle concentration differences were comparable for $NO_2$ and $O_3$ (Figure 6c and 6d), but mean $O_3$ concentrations exceeded mean $NO_2$ concentrations (Table 8).

During May 1 – 12, locations on the east side of San Francisco experienced higher $NO_2$ concentrations and lower $O_3$ concentrations than central and western locations (Figure 6). This result is consistent with typically prevailing winds from the west to northwest and with high traffic volumes on major freeways, I-80 (Bay Bridge), I-280, and US 101, which are expected

to yield higher emissions and ambient concentrations closer to areas with higher traffic volumes. Because fresh NO emissions initially reduce ambient $O_3$ concentrations, $O_3$ concentrations are typically lower where $NO_2$ concentrations are higher. The results of this limited analysis indicate that the measurement system can reveal differences among air pollutant levels occurring in different neighborhoods during short (i.e., days to weeks) time periods.



The San Francisco results reveal mean 1-km scale enhancements (FAMD) in $NO_2$ and $O_3$ up to 117 % and 46 %, respectively,
during the two-week sampling period. The results obtained for 1-km averages can be further examined to demonstrate higher
variability on smaller spatial scales. We compared the standard deviations of the 1-km mean intervehicle $NO_2$ differences to
the corresponding 1-km mean $NO_2$ concentrations to characterize variability within 1-km spatial averages. These ratios
(standard deviation of intervehicle difference/mean concentration) ranged from 0.5 to 3.0 (average = 1.3). Within the 1-km
averages, therefore, vehicle-to-vehicle $NO_2$ concentration differences varied by factors of 1 – 6 (twice the standard deviation
of the mean differences) times the mean observed 1-km average concentrations.

Another indicator of spatial variability at finer resolution is the FMAD: as previously noted, the FMAD provides a measure of
the variability of individual measurements because it averages absolute values of concentrations and is therefore relevant to
understanding the comparability of high-resolution measurements. For the San Francisco data, the FMAD represents the
variability of the 1-minute time averages that comprise each 1-km spatial average. The average of the FMAD values across all
1-km spatial averages was 0.74, nearly twice as high as the average FAMD of 0.44.

### 3.5.2 San Joaquin Valley, November 2016

Over ten months, driving routes in the northern San Joaquin Valley were located within the cities of Tracy (2017 population
90,890), Stockton (320,554), Manteca (76,247), Merced (84,464), Modesto (215,080), and Turlock (72,879)
(https://www.cacities.org/Resources-Documents/About-Us/Careers/2017-City-Population-Rank.aspx, last access December
2, 2019) (Table 1). The initial drives occurred November 16 – 23, 2016 (Figure 7; see also Figures S11 – S15). Because the
destinations were located over 100 km from where the cars were parked overnight in the San Francisco parking garage, the
cars drove longer distances and sampled more non-urban roads (both rural and high-traffic volume interstates) each day than
they did in Los Angeles or San Francisco. The San Joaquin Valley car-to-car comparisons therefore provide insight into
variations on larger spatial scales (e.g., 10 – 100 km), which are of interest for understanding enhancements of urban over non-
urban pollutant concentrations as well as pollutant transport between cities or subregions.

Between November 16 and 23, 2016, the cars drove on non-urban roads and on city streets in Stockton, Manteca, and Modesto,
providing information on pollutant concentrations in Stockton relative to other portions of the northern San Joaquin Valley
and in the eastern half of the San Joaquin Valley compared with the western side (Table 11; Figures 7 and S11 – S15). For
each geographical pairing, pollutant enhancements varied by pollutant and date (Table 12; see Tables S1 – S4 for detailed
tabulations). For example, relative to sampling in both a rural area and near I-205 in Tracy, Stockton exhibited enhancements
of $NO_2$ concentrations and $PM_{0.3-0.5}$ counts on November 16 along with deficits of NO and $O_3$. Since mean $NO_x$ (NO + $NO_2$)
concentrations in Stockton (31.3 ppbv) did not differ from the rural route (31.8 ppbv) (Tables S1, S2), the Stockton – rural
differences in NO and $NO_2$ concentrations may have been related to atmospheric chemical reactions and air mass aging. On
November 23, the Stockton – highway comparison exhibited the opposite pattern to November 16: deficits of $NO_2$
concentrations and $PM_{0.3-0.5}$ c $L^{-1}$ along with enhancements of NO and $O_3$ (Table 12) compared to routes in Modesto (within 1
km of Highway 99) and along Highway 99 (Modesto to Merced), Highway 140 (Highway 99 to I-5), and I-5 (Figures S15).





High traffic volumes are typical of Highway 99, so the results on this date indicate higher pollutant concentrations on and near major highways than on city streets in Stockton and in Modesto (Tables 12, S1).

The spatial analyses do not show consistent enhancements of pollutant concentrations in northern San Joaquin Valley cities

over concentrations occurring in surrounding areas. This result suggests a complex situation in which pollutant levels in the study cities depend on both local emissions and intra-regional pollutant transport. Similarly, the relationships between measured concentrations and intervehicle distance in the San Joaquin Valley depend upon the locations of the vehicles (Figure S16). Results for November 16 are shown for multiple species in Figure S17. $NO_2$ and particle numbers exhibited FAMDs exceeding 0.2 over most intervehicle distances. The largest FAMDs for $NO_2$ and particle numbers were associated with

contrasts between locations within the San Joaquin Valley and locations along an upwind boundary; these contrasts appear as intervehicle distances of 50 – 80 km, corresponding to times when Coltrane traversed the highway between San Jose (hour 11) and Crows Landing (near hour 14 at I-5 in the San Joaquin Valley) while Flora was sampling city streets in Stockton (Figure 7). Paired $O_3$ values were similar (FAMD < 0.2 up to intervehicle distances of 50 km), illustrating the regional character of $O_3$ in much of the northern San Joaquin Valley. The smaller FAMDs at 25 and 45 km intervehicle distances occurred when both

vehicles were sampling freeway locations in the urban San Francisco Bay area (Figure S17). The larger FAMDs at intervehicle distances of 15 km occurred when the cars traversed I-580 between Manteca and Hayward (near Castro Valley Freeway, Figure 7) on their return trip in the afternoon and the vehicles experienced differences in traffic levels due to their positions in urbanized versus nonurban portions of I-580 (hour 15, Figures 7, S17).

### 3.6 How Spatially Representative are Measurements from Regulatory Monitors?

Comparisons of mobile-platform concentrations to concentrations recorded by the downtown Los Angeles stationary monitor (CELA) showed that the FAMD for NO largely exceeded 1 (100 %); the $NO_2$ and $O_3$ FAMDs were less than 0.5 (50 %) and 0.2 (20 %), respectively, at car-monitor distances ranging from 1 to 7 km. The results indicate that the U.S. EPA classification of the downtown Los Angeles location as a neighborhood scale site (0.5 – 4 km zone of representation, Table 3) is appropriate for $NO_2$ and $O_3$. Comparisons of mobile monitors to data from the west Los Angeles monitor (WSLA) showed that the mobile

platforms recorded much higher concentrations of NO and $NO_2$ than the monitor at vehicle-to-monitor distances ranging from < 0.5 km to 5 km; for NO and $NO_2$, the FAMD exceeded 1.5 (150 %) and 0.6 (60 %), respectively. The results support the U.S. EPA classification of WSLA as a middle scale site (100 m to 0.5 km zone of representation, Table 3). The methods used for evaluating the spatial representativeness of CELA and WSLA are readily applied to other locations.

### 3.7 How Effective Were the Driving Routes for Addressing Study Questions?

The driving routes that were followed in this study were intended to address various research questions focused on evaluating mobile platform performance and spatial scales of representativeness (per previous subheadings in "Results and Discussion"). Different routes were deployed for different questions. The routes utilized in the comparisons with stationary regulatory





monitors in Los Angeles provided effective coverage of neighborhoods located 100 m to 4 km from two stationary monitors. The results supported the EPA classifications of those monitors.

The sampling conducted in San Francisco was intended to delineate spatial variations of pollutant concentrations across the city. Sampling during a single two-week period, which covered a subset of a compact urban environment, clearly revealed 300 m – 1 km spatial differences in pollution concentrations but varied by pollutant. In contrast, sampling was conducted over a much larger area in the northern San Joaquin Valley and the results were difficult to interpret from a limited (two-week) set of measurements because the spatial domains sampled were different on different days. For example, contrasts between an urban

area (Stockton) and areas surrounding Stockton were expected to yield information on the urban pollution enhancement in Stockton. However, three different types of environments were sampled in conjunction with the initial two weeks of Stockton measurements: (1) nearby cities (e.g., Manteca, Tracy, and Modesto, located 19 to 45 km from Stockton), (2) a major freeway (Highway 99, mean distance 61 km from Stockton), and (3) a rural area (56 km from Stockton). Establishing quantitative contrasts for each of these comparisons likely requires at least two weeks of data for each type of comparison (e.g., Stockton

vs rural). Such comparisons could be explored using the full San Joaquin Valley data set.

## 4 Conclusions

The Aclima, Inc. mobile measurement and data acquisition platform, which equips Google Street View cars with research-grade instruments to measure air quality at high spatial resolution, is an effective approach to obtaining improved understanding of spatial variations in air pollutant concentrations. Data provided by the system will be highly useful for

evaluating air quality management policies intended to reduce human air pollutant exposure, acute and chronic health impacts, and premature mortality. Audit results demonstrate that reference instruments in stationary vehicles are capable of reliably measuring NO, $NO_2$, $O_3$, and PN with bias and precision ranging from <5 % to <25 % at 1-s time resolution.

During experiments conducted in Los Angeles, San Francisco, and the San Joaquin Valley, California, collocated parked and moving mobile platforms replicated mean NO, $NO_2$, $O_3$ concentrations with mean differences in 1-s measurements ranging

from 0.2 to 5.6 ppbv; mean differences in $PN_{0.3\ to\ 0.5}$ varied from 500 to 21,000 c L$^{-1}$. On a relative basis, the mean differences between replicate mobile platforms ranged from 1 % to 37 % of the mean NO, $NO_2$, and $O_3$ concentrations and 2 % to 32 % of PN, with higher mean differences observed in the larger particle size ranges (which also had few numbers of particles). The majority (21 of 26) comparisons of collocated mobile platforms exhibited differences <20 % of the mean concentrations, thereby suggesting that differences exceeding 20 % obtained by vehicles operating simultaneously in different neighborhoods

represented measurable spatial variation.

Paired time-synchronous mobile measurements were used to characterize the spatial scales of concentration variations when vehicles were separated by <1 to 10 km. Measurements made in Los Angeles during August 2016 exhibited intervehicle FAMD that ranged from 2 % at 0.125 km to 14 – 44 % at 4.5 – 5.5 km (mean 12 %) for $NO_2$ and from 0.6 % at 0.125 km to 0 – 7 % at 4.5 – 5.5 km (mean 2 %) for $O_3$. The standard deviations of bin averages indicated that finer-scale (e.g., 100 – 300 m, 1-





minute averages) intervehicle variations were larger, indicating variability by up to a factor of two for $NO_2$ and a factor of six for NO (two standard deviations) within the binned intervehicle averages.

For NO and $PN_{0.3-0.5}$, bin-average mean differences exceeded 20 % for the same driving routes, indicating measured spatial variability exceeding the uncertainties in measurement methods when employing the mobile platforms. For NO, the standard deviations of bin averages ranged from 1.2 to 4.0 (average = 2.8), indicating that vehicle-to-vehicle NO concentration
differences varied by up to a factor of six (two standard deviations) within the binned intervehicle averages.

A data analysis approach was developed to characterize spatial variations in a manner that limits the confounding influence of diurnal variability. The approach involved examining synchronous differences between 1-minute measurements made by two mobile platforms, which were then averaged to one-kilometer resolution. The approach was illustrated using data from San Francisco, revealing mean 1-km scale enhancements in $NO_2$ and $O_3$ concentrations up to 117 % and 46 %, respectively, during
a two-week sampling period. Within the 1-km averages, vehicle-to-vehicle $NO_2$ concentration differences varied by factors of 1 – 6 times the mean observed 1-km average concentrations, implying higher variability at spatial scales <1 km (i.e., among 1-minute averages, corresponding to ~300 m distances). Locations on the east side of San Francisco experienced higher $NO_2$ concentrations and lower $O_3$ concentrations than central and western locations likely due to differences in traffic density and to meteorological factors, with prevailing winds from the west or northwest.
The mobile data were also used to provide insight into the spatial representativeness of measurements made at stationary monitoring locations. Comparisons of mobile measurements to data from two stationary monitors in Los Angeles indicate that the U.S. EPA classifications of the monitors as representative of neighborhood (0.5 – 4 km) or middle (100 m – 0.5 km) scale pollutant concentrations are appropriate. The methods used for evaluating the spatial representativeness of the two monitors are readily applied to other locations.

**5 Data Availability**

Access to Aclima QD2 data is provided by Google, Inc. on request through the Google Cloud Platform using Google Cloud Shell and Google Big Query (https://bigquery.cloud.google.com/table/street-view-air-quality:California_201605_201709_GoogleAclimaAQ.California_2016_2017?tab=details&pli=1).

**6 Contributions**

All authors contributed to the manuscript. P. S. and D. V. established science questions to be addressed in consultation with Aclima, Inc. and EPA staff. P.S coordinated the project for EPA through December 2018 and continued to work on the project after leaving EPA and while currently serving as a consultant to Aclima Inc. M. L. and B. L. managed the project for Aclima, Inc., supervised driving routes, evaluated measurement accuracy, and compiled data sets. C. B. and S. S. carried out analyses of the data sets. C.B. wrote the manuscript with contributions from each co-author.



## 7 Competing Interests

The authors declare that they have no conflict of interest. M. L. and B. L. are employed by Aclima, Inc. P.S. serves as a consultant to Aclima, Inc.

## 8 Disclaimer

Sections or the underlying technologies described herein are proprietary, owned by and subject to the intellectual property rights of Aclima, Inc. All rights reserved.

## 9 Acknowledgements

Funding for this project was provided by the EPA through a cooperative agreement with the Electric Power Research Institute (Coop No. 83925001-0). In-kind support was provided by Aclima and EPRI. Aclima collected the data, provided quality assurance, and provided access to the data used in this study. We thank Surender Kaushik who became the project manager after Paul Solomon retired from EPA. We also gratefully acknowledge K. Tuxen-Bettman, D. Herzl, O. Puryear, the Aclima mobile platform team, and the Google Street View team and drivers for their contributions to the project.

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



**Table 1. Summary of driving dates and plans.**

| Location | Dates | Driving plan |
|---|---|---|
| San Francisco | May – Sept 2016<br>April – June 2017 | Map every street in San Francisco, targeted driving |
| Los Angeles | Aug – Oct 2016 | Map specific neighborhoods with repeat visits |
| San Joaquin Valley | Nov 2016 – Apr 2017<br>June – Sept 2017 | Map multiple cities (Tracy, Stockton, Manteca, Merced, Modesto, Turlock), denser spatial coverage of Modesto |


**Table 2. Data sets used to evaluate spatial variability and to address individual research questions, including measurement uncertainty.**

| Location | Dates | Data analyses |
|---|---|---|
| San Francisco | May 1 – 31, 2017 | Stationary vehicle collocated comparisons (side-by-side parking-garage car measurements); neighborhood spatial variability |
| Los Angeles | August 3 – 12, 2016 | Stationary (side-by-side parking-garage) and moving vehicle collocated comparisons; neighborhood spatial variability; SCAQMD measurement audits |
| Los Angeles | September 20, 2016 | Comparisons to stationary-monitor data; SCAQMD measurement audits |
| San Joaquin Valley | Nov 16 – 23, 2016 | Stationary (side-by-side parking-garage) and moving vehicle collocated comparisons; urban-rural and interurban contrasts |




**Table 3. Sampling locations and dates for calibrations and audits through sample inlets conducted adjacent to stationary air quality monitors in Los Angeles.**

| Monitoring Site | Latitude | Longitude | Date |
|---|---|---|---|
| Long Beach near-road site (NRS) (W710) | 33.86266 | -118.19946 | 8/12/2016; 8/26/2016 |
| Los Angeles International Airport (LAXH) | 33.95500 | -118.43028 | 9/20/2016 |
| Santa Clarita | 34.38342 | -118.52822 | 10/6/2016; 10/25/2016 |

**Table 4. Stationary monitoring sites in Los Angeles for which the SCAQMD provided high-resolution (1-minute) measurements. Hourly-average gas and $PM_{2.5}$ mass concentrations are available for other locations through EPA public data archives.**

| Code | Name | Latitude | Longitude | 1-Minute Data | Scale[1] |
|---|---|---|---|---|---|
| CELA | Los Angeles N Main St[4] | 34.0664 | -118.2267 | CO, NO, $NO_2$, $O_3$ | Neighborhood |
| CMPT | Compton | 33.9014 | -118.2050 | CO, NO, $NO_2$, $O_3$ | Multiple[2] |
| HDSN | Long Beach (Hudson) | 33.8022 | -118.2197 | CO, NO, $NO_2$, $O_3$ | Neighborhood |
| LAXH | LAX-Hastings | 33.9550 | -118.4303 | CO, NO, $NO_2$ | Neighborhood |
| SLBH | South Long Beach[4] | 33.7922 | -118.1753 | | Neighborhood |
| W710 | Long Beach Route 710 | 33.8594 | -118.2003 | $PM_{2.5}$ mass | Micro |
| WSLA | Los Angeles-VA Hospital | 34.0508 | -118.4564 | CO, NO, $NO_2$, $O_3$ | Multiple[3] |

[1] Neighborhood scale = 0.5 km to 4 km; middle scale = 100 m to 0.5 km; micro scale = several meters to ~100 m

[2] Neighborhood scale for $O_3$; middle scale for other species

[3] Middle scale for $NO_2$; neighborhood scale for $O_3$

[4] Hourly $PM_{2.5}$ or $PM_{10}$ measurements available





**Table 5. Performance summary of the gas-phase instruments (NO, NO₂, O₃, and CH₄) in parked vehicles (Lunden and LaFranchi, 2017).**

| Pollutant (Car) | Bias (ppbv)[1] | Precision (ppbv)[1] | Limit of Detection[2] (2σ, 1 sec) (ppbv) |
|---|---|---|---|
| NO (Coltrane) | ± 2.1% + 0.3 | ± 2.3% ± 0.3 | 1.5 |
| NO (Flora) | ± 3.6% + 0.3 | ± 4.3% ± 0.3 | 1.7 |
| NO₂ (Coltrane) | ± 2.1% ± 0.4 | ± 2.8% ± 0.5 | <0.1 |
| NO₂ (Flora) | -2.4% + 0.2 | ± 2.2% ± 0.2 | <0.1 |
| O₃ (Coltrane) | ± 2.1% ± 0.5 | ± 2.4% ± 0.6 | 1.8 |
| O₃ (Flora) | ± 2.0% ± 0.4 | ± 2.3% ± 0.5 | 1.8 |
| CH₄ (Coltrane) | ± 3.3 | ± 3.3 | n/a |

[1] Bias and precision are expressed as the upper bounds (at 90% confidence) of bias and precision metrics determined from differences between measured and target (audit) concentrations (Camalier et al., 2007).

[2] Limit of detection (LOD) is defined as the minimum concentration at which an observation can be discriminated from zero (with 95% confidence) at the specified sampling frequency (2 standard deviations of zero gas measurements).




**Table 6. Performance of particle instruments (PN and BC) based on collocated parked vehicles. Evaluations performed between May 2016 and August 2017 (Lunden and LaFranchi, 2017).**

| Pollutant | Bias[1] | Precision[2] | RMSE[3] |
|---|---|---|---|
| $PN_{0.3 - 0.5}$ | ± 10.9% | ± 9.8% | 1293 c L$^{-1}$ (1 sec) |
| | | | 920 c L$^{-1}$ (1 min) |
| $PN_{0.5 - 0.7}$ | ± 7.2% | ± 7.5% | 471 c L$^{-1}$ (1 sec) |
| | | | 237 c L$^{-1}$ (1 min) |
| $PN_{0.7 - 1.0}$ | ± 11.3% | ± 10.0% | 170 c L$^{-1}$ (1 sec) |
| | | | 46 c L$^{-1}$ (1 min) |
| $PN_{1.0 - 1.5}$ | + 25.7% | ± 13.2% | 69 c L$^{-1}$ (1 sec) |
| | | | 9 c L$^{-1}$ (1 min) |
| $PN_{1.5 - 2.5}$ | + 25.7% | ± 15.6% | 71 c L$^{-1}$ (1 sec) |
| | | | 10 c L$^{-1}$ (1 min) |
| BC | ±11.9% ± 0.07 µg m$^{-3}$ | not estimated | ± 27.3% ± 0.26 µg m$^{-3}$ (1 sec) |
| | | | ± 15.6% ± 0.08 µg m$^{-3}$ (10 sec) |
| | | | ± 11.1% ± 0.05 µg m$^{-3}$ (1 min) |

[1] Bias for PN is calculated according to Camalier et al. (2007) where the values obtained by one car (Car A) are substituted for target (audit) concentrations. The positive sign of the bias estimate for the $PN_{1.0-2.5}$ (c L$^{-1}$) indicates a tendency of one instrument (Car B) to be biased high relative to the other instrument (Car A). Because BC concentrations were often close to LOD, bias for BC was estimated from linear least squares regression of bias vs concentration. A single bias value was estimated for each 6-hour collocation period using 1-minute aggregations from two vehicles. The bias estimates were regressed against the mean concentrations measured for the corresponding times. The relative and absolute components of bias were identified from the slope and intercept, respectively, of this linear regression ($r^2 = 0.37$, p-value < 0.0001).

[2] Precision is calculated according to Camalier et al. (2007) where the mean concentrations obtained by two cars are substituted for target (audit) concentrations.

[3] PN RMSE is determined from the vehicles' PN concentration differences relative to the means of the PN measured by the vehicles. RMSE for BC is estimated through a linear regression method (RMSE vs concentration) analogous to the procedure for estimating BC bias.





**Table 7. Mean ambient concentrations and sample sizes as measured by the mobile platforms in each of the example study areas.[1]**

| Subset | NO (ppbv) | NO$_2$ (ppbv) | O$_3$ (ppbv) | CH$_4$ (ppmv) | PN$_{0.3-0.5}$ (c L$^{-1}$)$^2$ | PN$_{0.5-0.7}$ (c L$^{-1}$)$^2$ | PN$_{0.7-1.0}$ (c L$^{-1}$)$^2$ | PN$_{1.0-1.5}$ (c L$^{-1}$)$^2$ | PN$_{1.5-2.5}$ (c L$^{-1}$)$^2$ | PN$_{>2.5}$ (c L$^{-1}$)$^2$ |
|---|---|---|---|---|---|---|---|---|---|---|
| LA1[3] | 10.5 | 15.3 | 44.1 | NA | 82,209 | 6725 | 1437 | 600 | 680 | 172 |
| | 589,555 | 626,136 | 228,498 | NA | 338,033 | 338,033 | 338,033 | 338,033 | 338,033 | 338,033 |
| LA2[4] | 21.4 | 22.5 | 37.7 | 2.17 | 42,818 | 4403 | 1251 | 537 | 748 | 274 |
| | 889,010 | 909,722 | 377,183 | 524,128 | 620,421 | 620,421 | 620,421 | 620,421 | 620,421 | 620,421 |
| SJV1[5] | 17.0 | 17.9 | 23.3 | 2.04 | 22,050 | 2769 | 742 | 304 | 375 | 153 |
| | 478,671 | 766,946 | 143,796 | 279,863 | 572,851 | 572,851 | 572,851 | 572,851 | 572,851 | 572,851 |
| SJV2[6] | 10.2 | 13.6 | 28.9 | 1.98 | 11,933 | 2527 | 1015 | 418 | 451 | 151 |
| | 294,514 | 393,917 | 35,215 | 140,022 | 283,179 | 283,179 | 283,179 | 283,179 | 283,179 | 283,179 |
| SF[7] | 6.0 | 10.3 | 26.5 | 1.98 | 13,947 | 4934 | 2288 | 922 | 868 | 154 |
| | 738,089 | 793,318 | 372,470 | 418,704 | 579,802 | 579,802 | 579,802 | 579,802 | 579,802 | 552,739 |

[1] Sample sizes are total number of 1-sec measurements summed across vehicles. Means are weighted by the number of measurements per vehicle.

[2] Particle number in size fractions 0.3 – 0.5 μm, 0.5 – 0.7 μm, 0.7 – 1.0 μm, 1.0 – 1.5 μm, 1.5 – 2.5 μm, > 2.5 μm.

[3] LA1 = Los Angeles, August 3 – 12, 2016 (8 days). BC, CH$_4$ = 1 car; NO, O$_3$, NO$_2$, and PN = 2 cars.

[4] LA2 = Los Angeles, September 12 – 30, 2016 (14 days). BC, CH$_4$ = 1 car; NO, O$_3$, NO$_2$, and PN = 2 cars.

[5] SJV1 = San Joaquin Valley, November 16 – 23, 2016 (6 days). BC, CH$_4$ = 1 car, NO and O$_3$ = 2 cars, NO$_2$ and PN = 3 cars.

[6] SJV2 = San Joaquin Valley, March 20 – 29, 2017 (6 days). BC, CH$_4$ = 1 car, NO and O$_3$ = 2 cars, NO$_2$ and PN = 2 cars.

[7] SF = San Francisco, May 1 – 12, 2017 (10 days). BC, CH$_4$ = 1 car; NO, O$_3$, NO$_2$, and PN = 2 cars.




**Table 8. External calibration checks (zero and span) performed in Los Angeles with equipment and gas standards managed by the SCAQMD compared with internal checks performed by Aclima one month prior to the Los Angeles deployment, one month following this deployment, and during a 1-week return to San Francisco in the middle of the deployment. External and Aclima calibration checks were conducted through the inlet lines of the mobile platforms.**

| Species | Audit | Bias (% ± ppbv) | Precision (% ± ppbv) | Number of Span Checks | Number of Zero Checks |
|---------|-------|-----------------|----------------------|-----------------------|-----------------------|
| NO | Aclima | ± 3.5% + (< 1) | ± 4.5% + (< 1) | 22 | 22 |
| | SCAQMD | ± 8.2% + (< 1) | ± 6.0% + (< 1) | 10 | 10 |
| NO$_2$ | Aclima | - 3.7% ± 0.4[1] | ± 3.7% ± 0.4 | 19 | 20 |
| | SCAQMD | - 1.9% ± 0.6[1] | ± 4.9% ± 0.6 | 6 | 10 |
| O$_3$ | Aclima | ± 2.4% ± 0.9 | ± 2.3% ± 1.1 | 20 | 18 |
| | SCAQMD | ± 3.3% ± 1.2 | ± 3.8% ± 1.5 | 10 | 10 |

[1] Negative bias only



**Table 9. Performance summary for measurements reported by collocated vehicles (mean difference ± 1 standard deviation; mean concentrations in parentheses). Standard deviations are reported here to indicate the variability of the 1-s differences. Mean differences provide a measure of average intervehicle differences. For periods when three vehicles were driven, the largest mean difference between vehicles is listed. The signs of the mean differences are not indicated because no vehicle is an audit standard. All values were determined from 1-s time resolution data.**

| Setting | Period[1] | NO[2] (ppbv) | $O_3$[2] (ppbv) | $NO_2$[2] (ppbv) | $PN_{0.3-0.5}$[2] (c $L^{-1}$) |
|---|---|---|---|---|---|
| Parking structure[3] | LA1 | 0.6 ± 49.5 (11.3) | 1.5 ± 8.1 (41.6) | 0.3 ± 12.0 (15.8) | 18346 ± 21024 (81929) |
| Parking structure[3] | LA2 | 3.9 ± 66.9 (21.5) | 1.0 ± 9.9 (34.6) | 1.9 ± 14.3 (22.7) | 6525 ± 20049 (44058) |
| Parking structure[3] | SJV1 | 0.5 ± 2.1 (3.8) | 4.5 ± 2.3 (18.9) | 1.0 ± 1.5 (16.7) | 1126 ± 3922 (12527) |
| Parking structure[3] | SF | 0.2 ± 7.5 (3.5) | 0.8 ± 3.9 (24.2) | 1.1 ± 5.7 (6.2) | 507 ± 1865 (14154) |
| Moving[4], < 10 m | SJV1[8] | NA[10] | NA[10] | 5.6 ± 32.7 (15.1) | 132 ± 4242 (7661) |
| Moving[5], 10–100 m | SJV1[8] | NA[10] | NA[10] | 1.9 ± 20.1 (16.2) | 454 ± 2478 (5883) |
| Moving[6], < 10 m | SF-LA[9] | 13.8 ± 56.7 (27.9) | 1.8 ± 2.2 (40.9) | 3.4 ± 9.4 (16.8) | 20797 ± 5410 (64187) |
| Moving[7], 10–100 m | SF-LA[9] | 5.1 ± 49.1 (26.5) | 0.5 ± 3.2 (42.2) | 1.0 ± 12.1 (17.3) | 19294 ± 7670 (60046) |

[1] LA1 = August 3 – 12, 2016 (8 days); LA2 = September 12 – 30, 2016 (14 days); SJV1 = November 16 – 23, 2016 (6 days); SJV2 = March 21 – 30, 2017 (6 days)

[2] Vehicle-to-vehicle concentration differences were determined from 1-s measurements. Means and standard deviations of paired differences were determined for each data pair. Time periods when a vehicle was sampling through a calibration port (whether a calibration was in process) were excluded to ensure that vehicles were sampling the same ambient air for all

comparisons.

[3] One parking structure is in Los Angeles and was used for LA1 and LA2. The second parking structure is in San Francisco and was used for all SF and SJV drives.

[4] intervehicle distance < 10 m (average = 5 m), average speed = 3.0 m $s^{-1}$ (10.6 km h $^{-1}$)

[5] intervehicle distance 10 – 100 m (average = 32 m), average speed = 25.6 m $s^{-1}$ (92.0 km h $^{-1}$)

[6] intervehicle distance < 10 m (average = 6 m), average speed = 5.9 m $s^{-1}$ (21.2 km h $^{-1}$)

[7] intervehicle distance 10 – 100 m (average = 44 m), average speed = 27.7 m $s^{-1}$ (99.7 km h $^{-1}$)

[8] November 16, 2016 (I-580 and other locations, Flora and Rhodes, Figure S2)

[9] August 1, 2016 driving from San Francisco to Los Angeles (I-5 and other locations)

[10] Not available. SJV1, one car (Rhodes) of collocated moving pair lacked NO and $O_3$ samplers.




**Table 10. Comparison of mobile-platform to collocated stationary-site measurements made at LAXH on September 20, 2016. The two cars alternated positions between an audit location 6.6 m for Coltrane and 8.5 m for Flora horizontal distance from the ground-level coordinates of the LAXH monitor (inlet situated 4.2 m agl inside a fenced enclosure) and a sampling location further from the monitor (24.1 m for Coltrane and 18.5 m for Flora). Data from the audit tests are excluded. The Coltrane audit period was 10:22 a.m. – 12:20 p.m. PDT (n = 119). The Flora audit period was 9:19 a.m. – 10:20 p.m. PDT (n = 56). The means ± standard errors of the means were determined for each car from the 1-minute measurements made at the two distances from the stationary monitor. Standard errors indicate the uncertainties of the mean concentrations and mean differences. Differences of 1-minute measurements were determined prior to averaging. The variabilities of the 1-minute differences can be obtained by multiplying standard errors by square root of sample size (n).**

| Platform | $N^1$ | NO (ppbv) | $NO_2$ (ppbv) | $O_3$ (ppbv) | $O_x$ (ppbv)[2] |
|---|---|---|---|---|---|
| Coltrane | 56 | 17.7 ± 1.1 | 37.7 ± 1.8 | 16.2 ± 0.6 | 53.9 ± 1.3 |
| Flora | 56 | 18.8 ± 1.2 | 37.0 ± 1.7 | ND | ND |
| LAXH | 56 | 19.0 ± 1.2 | 36.6 ± 1.7 | 20.3 ± 0.4 | 56.9 ± 1.3 |
| Coltrane - LAXH | 56 | -1.3 ± 0.4 | 1.2 ± 0.4 | -4.1 ± 0.3 | -3.0 ± 0.2 |
| Flora - LAXH | 56 | -0.2 ± 0.3 | 0.6 ± 0.3 | ND | ND |
| Coltrane | 119 | 4.5 ± 0.4 | 16.5 ± 1.0 | 41.7 ± 0.4 | 52.2 ± 0.6 |
| Flora | 119 | 3.1 ± 0.3 | 14.7 ± 0.9 | 36.1 ± 0.7 | 50.7 ± 0.3 |
| LAXH | 119 | 4.1 ± 0.4 | 14.6 ± 0.9 | 38.8 ± 0.7 | 53.4 ± 0.3 |
| Coltrane - LAXH | 119 | -0.1 ± 0.2 | 0.3 ± 0.2 | -0.9 ± 0.3 | -0.2 ± 0.3 |
| Flora - LAXH | 119 | -1.0 ± 0.1 | 0.04 ± 0.2 | -2.6 ± 0.2 | -2.6 ± 0.2 |

[1] Total minutes. Flora audit period 9:19 a.m. – 10:20 p.m. PDT (n = 56) and Coltrane audit period 10:22 a.m. – 12:20 p.m. PDT (n = 119). Sample sizes for individual measurements may be smaller due to excluding audit values. Mean paired differences are computed only for non-audit samples.

[2] $O_x = NO_2 + O_3$





**Table 11. Dates, locations, and times when vehicle pairs sampled different areas within the northern San Joaquin Valley.**

| Date | Areas Sampled | Vehicles | Hours | Mean Distance (km) | Species Measured by Both Vehicles |
|---|---|---|---|---|---|
| Nov 16 | Stockton – Rural | Flora – Coltrane | 12 – 14 | 56.2 | NO $NO_2$ $O_3$ PM |
| Nov 16 | Stockton – Tracy | Flora – Rhodes | 12 – 14 | 37.5 | $NO_2$ PM |
| Nov 17 | Stockton – Manteca | Coltrane – Flora | 13 – 14 | 18.7 | NO $NO_2$ $O_3$ PM |
| Nov 17 | Stockton – Stockton | Coltrane – Rhodes | 13 – 14 | 1.2 | $NO_2$ PM |
| Nov 17 | Stockton – Manteca | Rhodes – Flora | 12 – 15 | 17.9 | $NO_2$ PM |
| Nov 18 | East – West SJV | Flora – Coltrane | 12 – 14 | 49.9 | NO $NO_2$ $O_3$ PM |
| Nov 18 | East – West SJV | Rhodes – Coltrane | 12 – 14 | 49.7 | $NO_2$ PM |
| Nov 21 | East – West SJV | Flora – Rhodes | 12 – 14 | 47.1 | $NO_2$ PM |
| Nov 21 | East – West SJV | Coltrane – Rhodes | 12 – 14 | 37.9 | $NO_2$ PM |
| Nov 22 | Stockton – Modesto | Flora – Coltrane | 12 – 14 | 43.9 | NO $NO_2$ $O_3$ PM |
| Nov 22 | Stockton – Modesto | Flora – Rhodes | 12 – 14 | 44.6 | $NO_2$ PM |
| Nov 23 | Stockton – Modesto | Flora – Rhodes | 10 – 13 | 30.4 | $NO_2$ PM |
| Nov 23 | Stockton – Highway | Flora – Coltrane | 10 – 13 | 61.0 | NO $NO_2$ $O_3$ PM |



**Table 12. Fractional mean differences (FMD) when vehicle pairs sampled different areas within the northern San Joaquin Valley. Vehicles A and B correspond to the first and second areas sampled, respectively. Uncertainties are one standard error of the means. NA = not available; one car (Rhodes, R) measured only NO₂ and PM concentrations.**

| Date | Areas Sampled | Car[1] A–B | NO FMD | NO$_2$ FMD | O$_3$ FMD | PM FMD |
|---|---|---|---|---|---|---|
| Nov 16 | Stockton – Rural | F–C | -0.30 ± 0.02 | 0.30 ± 0.02 | -0.24 ± 0.004 | 0.96 ± 0.01 |
| Nov 16 | Stockton – Tracy | F–R | NA | 0.46 ± 0.02 | NA | 0.14 ± 0.01 |
| Nov 17 | Stockton – Manteca | C–F | 0.61 ± 0.03 | -0.16 ± 0.01 | 0.01 ± 0.004 | 0.02 ± 0.004 |
| Nov 17 | Stockton – Stockton | C–R | NA | 0.007 ± 0.01 | NA | 0.11 ± 0.003 |
| Nov 17 | Stockton – Manteca | R–F | NA | -0.18 ± 0.01 | NA | 0.12 ± 0.004 |
| Nov 18 | East – West SJV | F–C | -0.61 ± 0.05 | -0.30 ± 0.02 | NA | 0.23 ± 0.004 |
| Nov 18 | East – West SJV | R–C | NA | -0.23 ± 0.02 | NA | 0.14 ± 0.004 |
| Nov 21 | East – West SJV | F–R | NA | -0.30 ± 0.02 | NA | -0.13 ± 0.008 |
| Nov 21 | East – West SJV | C–R | NA | 0.30 ± 0.02 | NA | 0.23 ± 0.006 |
| Nov 22 | Stockton – Modesto | F–C | 0.36 ± 0.03 | 0.49 ± 0.01 | -0.42 ± 0.01 | 0.10 ± 0.005 |
| Nov 22 | Stockton – Modesto | F–R | NA | 0.70 ± 0.01 | NA | -0.12 ± 0.006 |
| Nov 23 | Stockton – Modesto | F–R | NA | -0.09 ± 0.01 | NA | -0.65 ± 0.02 |
| Nov 23 | Stockton – Highway | F–C | 0.40 ± 0.03 | -0.09 ± 0.01 | 0.18 ± 0.006 | -0.57 ± 0.02 |

[1] C = Coltrane, F = Flora, R = Rhodes





**Figure 1. Mean vehicle speeds and pollutant concentrations averaged by hour over all Los Angeles driving days between August 3 and 12, 2016. Standard errors of the means are plotted but are generally smaller than the symbol sizes.**



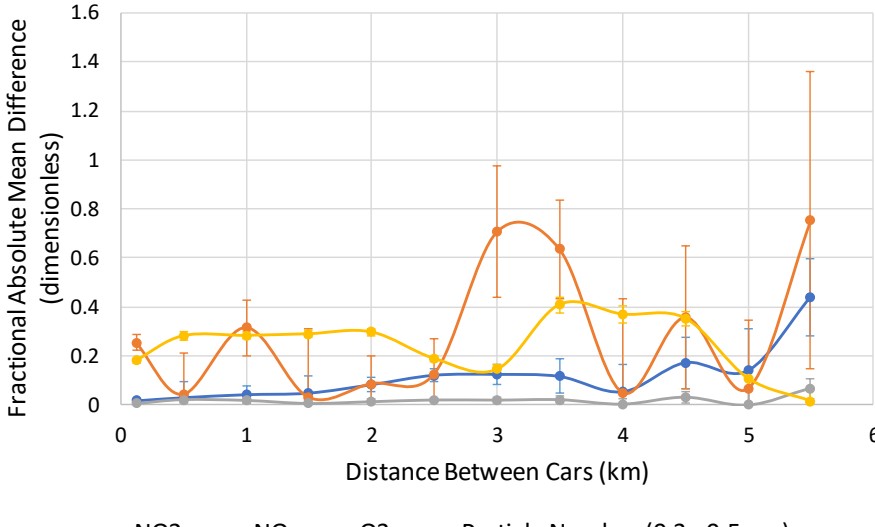

**Figure 2.** **Intervehicle FAMD vs mean intervehicle distance associated with sampling in Los Angeles (near CELA) from August 3 – 12, averaged over 0.5 km bins (0 – 0.25 km, 0.25 – 0.75 km, etc.). Error bars are 1-sigma uncertainties determined as described in the definition of FAMD. The sizes of the error bars reflect variations in the number of samples in each bin (N = 14 to 2433) as well as sampling variability.**










**Figure 3. Fractional absolute mean difference (FAMD) for (a) NO, (b) NO₂, and (c) O₃ vs mean intervehicle distance for August 3 – 12, 2016, Los Angeles sampling, averaged over 0.5 km bins. Error bars are 1 sigma uncertainties as described in the text. The sizes of the error bars reflect variations in the number of samples in each bin (N = 3 - 19 at 6.5 km to 222 – 338 at 3.5 km). The 3 km bin (N = 1273 – 3906) consists primarily of measurements made in the parking garage.**


**Figure 4. Mobile platform monitoring and WSLA measurements versus distance between cars and WSLA on four days (September 13, 19, 26, and 29, 2016) when the cars drove near WSLA. The first bin includes all distances less than 0.5 km; the minimum distance between cars and monitor was 158 m. Locations are indicated. Standard errors of the means are shown but most are smaller than the symbols.**







**Figure 5. FAMD between mobile platform monitoring and WSLA measurements versus distance between cars and WSLA on four days (September 13, 19, 26, and 29, 2016) when the cars drove near WSLA. The first bin includes all distances less than 0.5 km; the minimum distance between cars and monitor was 158 m. Locations are indicated. One-sigma uncertainties of the FAMD were determined as described in the definition of FAMD in the text.**





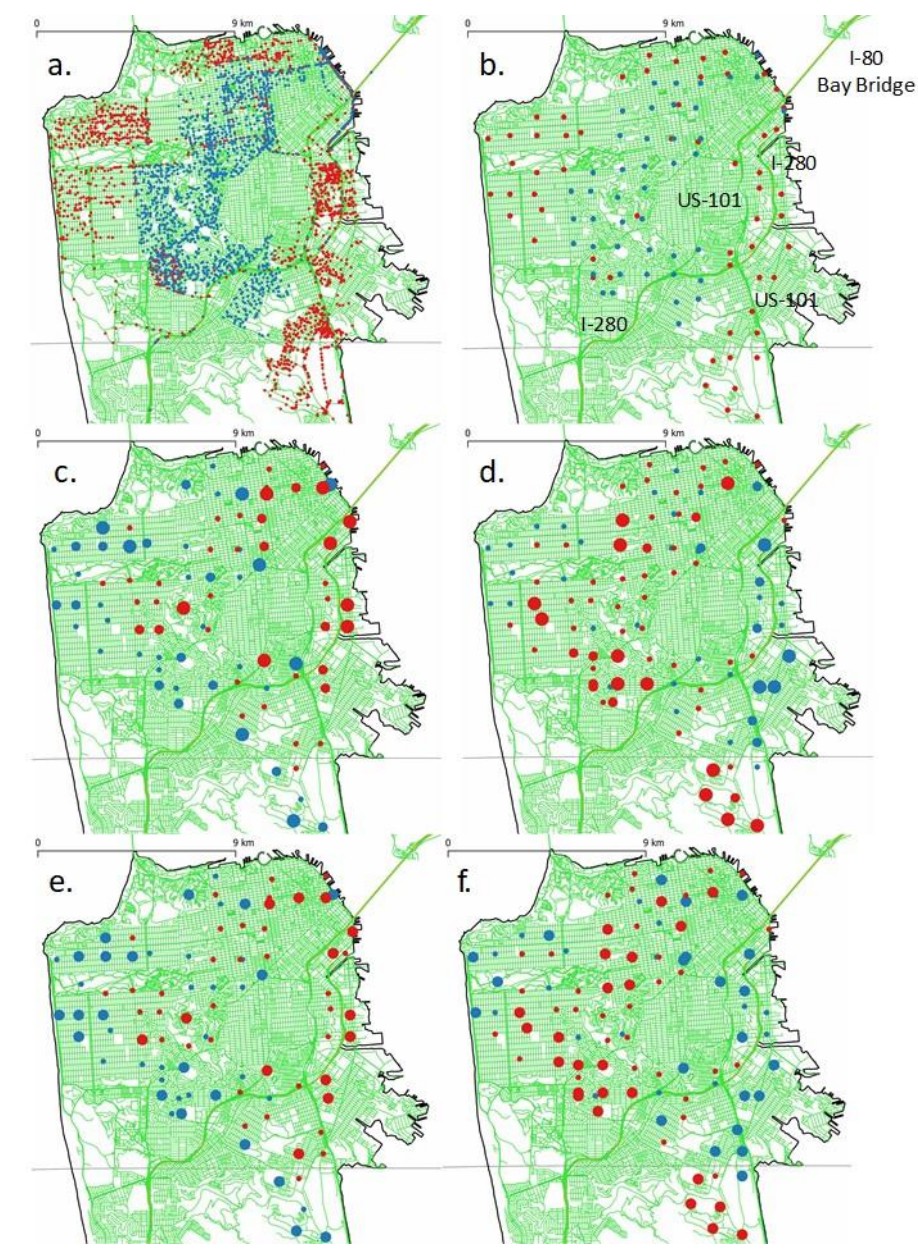

**Figure 6. San Francisco sampling locations and results for May 1 through 12, 2017: (a) 1-minute resolution locations (red = Flora, blue = Coltrane), (b) 1-kilometer resolution locations (red = Flora, blue = Coltrane), (c) NO₂ intervehicle differences (red = positive, blue = negative; small symbol = < -4 or > 4 ppbv, medium = -4 to -2 or 2 to 4 ppbv, large = -2 to +2 ppbv), (d) O₃ intervehicle differences (same scale as NO₂), (e) NO₂ FMD (red = positive, blue = negative; small symbol = < -0.5 or > 0.5, large = -0.5 to +0.5), (f) O₃ FMD (red = positive, blue = negative; small = < -0.05 or > 0.05, large = -0.05 to +0.05). Maps generated with QGIS version 3.2.2 (https://qgis.org/en/site/) open-source software licensed under the GNU General Public License (http://www.gnu.org.licenses). California coastline shapefile obtained from the OpenStreetMap community (www.openstreetmap.org) and MapCruzin (www.mapcruzin.com), licensed under the Creative Commons Attribution Share-Alike 2.0 license. U.S. highways and California county boundary shapefiles obtained from U.S. Bureau of the Census TIGER/Line shapefiles public data (https://www.census.gov/geographies/mapping-files/time-series/geo/tiger-line-file.html).**

1010



**November 16, 2016**

Coltrane – red          Flora – blue          Rhodes - green

Locations of each car at beginning of each hour are shown by symbols

**Figure 7. San Joaquin Valley driving routes on Nov 16, 2016. The positions of each car at the beginning of each hour are marked. The drives began and ended at the parking garage in San Francisco. Locations of cities identified in the text are also shown. Map generated with QGIS version 3.2.2 (https://qgis.org/en/site/) open-source software licensed under the GNU General Public License (http://www.gnu.org.licenses). California coastline and state highway shapefiles obtained from the OpenStreetMap community (www.openstreetmap.org) and MapCruzin (www.mapcruzin.com), licensed under the Creative Commons Attribution Share-Alike 2.0 license. U.S. highways and California county boundary shapefiles obtained from U.S. Bureau of the Census TIGER/Line shapefiles public data (https://www.census.gov/geographies/mapping-files/time-series/geo/tiger-line-file.html).**