# Peer review of "This supplement includes maps of monitoring sites and driving routes, data tabulations, and supplemental data analyses."

_Atmospheric Measurement Techniques, 2019_

## Referee Comment (RC1) · Anonymous Referee #1 · 26 Feb 2020

General Comments It is clear that a lot of work and effort was put into this manuscript and the authors did a nice job explaining the statistical tests used to assess data comparability. I especially appreciate the documentation of instrument bias and precision and how that was used to determine when spatial differences were present vs. likely not. While it is the opinion of this referee that additional information and clarifications are needed (see Specific Comments), the authors have put forth a worthwhile method to assess spatial representativeness of air pollution data and the scientific community

would benefit from having this manuscript published.

Specific Comments 1. Methods: some information on the cars should be included, not just referenced in another paper. Were the cars' engines running while parked (e.g., in the garage, near stationary monitors, etc.)? What was used to power the instruments? 2. It is great that the inlets were designed to minimize self-sampling, but were additional steps taken during post-processing to remove potential periods of self sampling, or of sampling the Google car in front? 3. It would be good to document the limitations of the study (e.g., no overnight monitoring on roads or in early morning when the boundary layer is likely at its lowest). 4. Table 1 - was winter included in the San Joaquin Valley measurements (Nov '16-Apr '17) or was it just fall and spring? 5. Section 2.2 - were the cars parked on the roof of the parking structures or on a lower level? Depending, this could explain why GPS uncertainties were not comparable to manufacturer specs at times. Tall buildings nearby (if present) would also impact GPS performance. 6. Lines 286-289: the closely-spaced moving vehicle condition makes it highly likely that the following car is measuring exhaust emissions from the lead car. This point should be mentioned in the manuscript. Did you try to correct for this? Why not drive side-by-side (road permitting)? 7. Could you please provide a list or table in the SI with the manufacturer and model of all instruments used in the study along with response time and measurement frequency. Even if this information was referenced in another paper, it should be reported here. 8. Did you sync all instruments to the same time standard before measurements? Did you check instrument times at the completion of each day's measurement to quantify time drift? At measurement frequency of 1 Hz, time drift can have a major impact on data comparison. These details should be included in the methods section. 9. By comparing 1-min averages, which I understand is important in order to maintain higher spatial resolution, how are you able to separate out the spatial trends due to differences in regional concentrations as opposed to differences in measurements due to some very localized conditions (e.g., driving behind a truck for a short period of time with one Google car but not the other over the same time period)? Would some other comparisons be more appropriate, such as a

60-second moving 5th percentile, or something comparable, to smooth out hyper-local concentrations? 10. I am struggling to understand why plotting the measurements against distance between either the cars or between car and stationary monitor is the best way to present the data. Had the cars been driving different routes than the ones presented, the plots would be completely different. The distance between the cars is not driving the differences observed, it is the difference in the environments of the two cars at any given time. For example, the cars could both be in heavy traffic at 50 km away from each other (thus mean differences in concentrations are low), then at 75 km distance one car is still in heavy traffic while the other is in a quiet neighborhood away from highways (thus mean differences in concentrations are high). For example, Figures S16 and S17 are interesting, but it would be more informative to provide information on where each of the cars are (e.g., land use, traffic conditions, major roadway, etc.) when FAMD is higher or lower irregardless of the distance between the cars. Are all points where the cars are X distance away from each other aggregated together even if the positions were discontinuous? If so, I do not know how one could interpret this plot. 11. Section 3.6 (lines 476-479): An FAMD of 0.5 seems high to conclude that a reference monitor is representative of a neighborhood scale area.

Technical Corrections 1. Figure S3: The figure overlaps text. 2. Line 265: Remove "the" before "Los Angeles". 3. Line 299: Please clarify if minimal traffic was observed adjacent to the site during measurements or not. As written it is not clear if expectations were met or not. 4. Results & Discussion: Consider not using questions for headings.

---

## Referee Comment (RC2) · Anonymous Referee #2 · 26 Feb 2020

General comments: This study aims to use a mobile platform for air quality measurements for characterizing spatial variations of air pollution within 15 urban areas in California, USA. They obtained data from the Aclima, Inc. mobile measurement and data acquisition platform used to equip Google Street View cars between May 2016 and September 2017 at very high temporal and spatial resolutions. The results demonstrate that the approach used for quantifying spatial variations of air pollutant concentrations over the measurement periods is working well. They focused on examining

measurement capabilities and developed statistical methods for analyzing the data. This manuscript demonstrates the capabilities of a fairly new instrument and clearly the authors put a lot of effort in the measurements and data analysis including spatial and temporal error and accuracy. The referee thinks this manuscript is well written and the scientific community will benefit from this type of study. The referee finds this manuscript to be a good fit to be published in AMT after addressing the comments below.

Specific Comments: 1. There is no discussion or comparison between Aclima instrumentation and capabilities to other sensors in the market (e.g. Purple Air), including technical and accuracy information. Have the authors done any comparison studies at similar times and locations to demonstrate Aclima outperforming other sensors? 2. This analysis provides information on mobile air quality monitoring in a certain environment. The measurements represent air quality in urban locations near roads and that covers certain points/line measurements yet does not create a continuous air quality map. 3. PN is measured by the Aclima platform for different size bins. It is not clear how this measurement is evaluated, as the EPA monitors particulate matter mass concentration? 4. It is not clear why the distance between cars is important in the discussion. 5. All the measurements have been done for periods of several weeks and there is no 'long-term' monitoring campaign presented (e.g >1 year) that captures, for example, seasonality. This limitation of measurements period should be addressed in the discussion. 6. A description of the climatology at the different measurement locations is missing (e.g. temp, RH, and wind profiles, built area, type of road, no. of cars etc.). That can help understand some of the results. 7. The authors should do a better job in stating the limitations of the Aclima platform in this study set and in general. 8. Did the authors consider validating their results with continuous modeled data (CMAQ)? Or satellite data? Technical Comments: 1. General: The referee strongly suggests not having question marks "?" in titles. 2. Line 81-82: distance from adjacent air quality validation monitor <4m. While in the abstract, line 25 it's mentioned that the distance is <9m ?

[Figure]

---

## Referee Comment (RC3) · Paul A. Solomon et al. · 16 Mar 2020

General Comments: ———————————— This manuscript evaluates statistical methods used to characterize spatial variation and representativeness of mobile and stationary monitor measurements for several air pollutants across multiple cities in California. The ground-level pollutants include both gases and particulates. This method evaluation paper will have implications for the spatial characterization of air quality measurements, which aligns with AMT. The manuscript in its current state is not ready for publication and will benefit significantly from additional minor work.

[Figure]

Specific Comments: ———————————————— Line 137: Do the monitoring stations use the Network Time Protocol? If not, address discrepancies this may cause in measurement comparisons. Line 164: What is the impact of wind and GPS location uncertainties on data collected while stationary? Line 173: Where are the reported variabilities of the paired differences shown? Line 17: The time frame is misleading. Should indicate a few intensive (i.e. week to monthlong) campaigns were performed between May 2016 and September 2017. Lines 20-21: Lifetime of NOǎ2 is hours and O3 is ∼20 days in the troposphere. Observing the diurnal cycle and weekday weekend trends may be more appropriate than looking at a fortnight. Line 22: In-situ instrument or research-grade instruments. I'm sure they mean their instrument package. Line 31: Percentages of what? Concentration deltas? Line 75: Concentration decay rate from a point source will be highly variable and based on several meteorological parameters. Line: 109-110: What are the limitations of overnight calibration when cars are parked next to each other? Lines 106,109/110, 113: The first lines seem to imply the mobile platform intercomparison was made overnight, 113 implies it may have been only a short period (5 min, 30 min), the SI from their Apte et al. indicates it was several hours overnight. Not sure about an intercomparison in a parking garage either, especially if it was during a time when vehicles were entering or leaving (cold starts vs operating temp emissions). Line 119: Was the audit the same as is done with FRM/FEM monitors via the National Performance Audit Program (NPAP)? Line 122: In table 4 there needs to be an explanation of the scales and how EPA initially established each sitting. Line 125: Suggested to add sentence or phrase to cover why the other stations were not used. Sec. 2.3: Are there multiple BC and CH4 instruments or just one that was moved between cars? I'm assuming this was done because of inlet restrictions. Sec. 2.3: Was CARB contacted to ensure BC and CH4 observations were not present at sites? Some EPA sites have but don't advertise these observations. Lines: 247-249: Comparing different regions during different time periods without a detailed study of the meteorology is misleading if talking about local or neighborhood scales. Here are the climatological winds near San Joaquin Valley for March and November using

data between 1973-2019 at Buchanan Field Airport in Concord, CA. Figure 1 shows the month of March may be experiencing inflow from the Chevron processing plant in Richmond and dust (Coarse mode, not reported) from Dutra Materials quarry in Mc-Nears Beach, while to a much lesser extent in November (Figure 2). Since the data is presented as mean concentrations during the sampling periods, I'd bet the baseline PN concentrations are different for the two months. Lines 251-252: The deployed optical particle counter provided five size ranges why report only the smallest, then reference a paper regarding a measured size bin that was not reported in the paper? Lines 246-262: I'm not sure this section is representative and should be included here and should likely be absorbed by the following sections. Line 265: Typo. . . .vehicles drove in the Los Angeles Line 270: Why was the mean relative difference between the two calibrations so high? An absolute difference of 5% NPAP would require corrective actions. The calibration gases and flow meters used should be traceable to NIST for re-evaluation. Sec. 3.2: When comparing inter-vehicle observations were the vehicles traveling the same route (i.e. following each other) or just driving the same neighborhoods and passing by each other? Line 294: Were the vehicles were running during the LAXH comparison or were the instruments moved to shelter power and the vehicle engines shutoff? Sec. 3.3: Last sentence of section, CH4 emissions from vehicles is extremely small (something like <0.2% of anthropogenic emissions) and the lifetime of NO very short. This statement needs a citation, or it needs to be removed. Sec 3.4: This section will have a very large dependence on meteorological parameters. Sec. 3.4.1: The airmasses the vehicles are sampling are potentially different. An intervehicle comparison could be made in time and latitude. As it is, the comparisons are meaningless because we know the location of any vehicle at any given time and one may be sampling south of the Santa Ana Freeway and the other sampling all three major N-S freeways in the area. The attached Figure 3 shows winds are between 9am and 5pm averaged over Aug 3-16. Line 362: Driving near as in right past along Dowlen Dr or within n meters? Wilshire Blvd is ∼200m as is Federal Ave. Line 392: What grid is used? Fig. 6: Needs legend, different colors for positive and negative intervehicle

differences and FMD differences not red/blue, which were used to identify specific vehicles in the same figure. Line 424: Enhancements based on what? FAMD is comparing observations at the same time, is the enhancement based on location as stated in the paragraph before or between May 1-12? Line 440: Routes for November 16th, 2016 are not in SI but referenced in text. Include Line 457: Are traffic count data available? Line 459: Enhancements compared to what, background? Line 529: Enhancements based on what? General: Overall distance bins should be the same for all missions. Seems like all the analysis times were weekday (do Google Street View vehicles drive on weekends)?

TECHNICAL CORRECTIONS: —————————————— Line 35: Suggested to add spatial variability context for pollutants to introduction as this has implications on reported uncertainties. Line 155: LOD is defined in Table 5 subtext, but not in text. Consider defining in main text. Lines 172-174: Suggested to remove 'merge' detail, as it seems superfluous to the reader, and combine the two sentences into one focusing on temporally coincident pairing. Lines 185, 190, 195, 200: 'Car B Difference' could be misleading. It is suggested to move the word 'Difference' to after the word 'Mean' (i.e., Mean Difference) or use wording such as 'Mean [Absolute] Difference between Car A and B' in the numerator. Line 206: Z is not defined. Line 216: MD already defined in line 185. Lines 211 and 222: Consistency in section references.

———————————————————

[Figure]

**Fig. 1.**

[Figure]

**Fig. 2.**

[Figure]

**Fig. 3.**

---

## Author Comment (AC1) · 24 Mar 2020

Thank-you for your reviews, which will help us improve the manuscript. We will incorporate responses to all questions and suggestions in a revised manuscript. We will also provide separate responses to each individual review.

In all three reviews, several questions occurred about two topics: measurement limitations and the statistical approach of differencing time-synchronized concentrations.

[Figure]

We will address these questions at appropriate places in the manuscript. We will also provide further context and explanation by expanding the brief overview description of the study and study objectives that appears at the end of the introduction (lines 78 – 100). We will add the following paragraph at line 77.

The mobile sampling discussed here and in Apte et al. (2017) is limited to weekdays between ∼9 a.m. and 5 p.m. Sampling is necessarily conducted along roads and streets. Depending on the number of repeated driving segments, vehicles sample different road segments on different days or at different times of day. These limitations are important considerations for studies whose goal is to develop pollutant maps that represent long-term concentration averages, and which are intended to correctly characterize spatial variations at a desired spatial scale. Our objectives are different, however. The principal objectives of our study are to examine the capabilities of research instruments when placed in stationary and moving vehicles, to compare our measurements with those obtained from stationary air quality monitors, to evaluate driving and sampling strategies, and to develop statistical methods that account for sampling limitations. Limitations that are specific to our study are that (1) it was conducted as a series of geographically separated sampling campaigns between May 2016 and September 2017, generally lacking the number of repeated driving routes previously used to generate pollution maps (Apte et al., 2017; Messier et al., 2018), and (2) no collection of driving routes completely covered any specific geographical domain (e.g., San Francisco or specific neighborhoods therein). The results presented here therefore focus on measurement and methodological questions that can be addressed with data available from the individual sampling campaigns. A set of research questions was developed initially and was then used to design the individual sampling campaigns. In analyzing the results, a need arose to distinguish between temporal variability (due, e.g., to sampling different places at different times) and spatial variability. Statistical methods were therefore developed to characterize spatial heterogeneity within and between neighborhoods by utilizing time-synchronized differences in the pollutant concentrations that were measured by different vehicles. Due to limited repeated sampling of individual

road segments, our estimates of spatial heterogeneity do not in themselves identify specific spatial coordinates of long-term high and low pollutant concentrations. However, areas with high spatial heterogeneity indicate where more intense future sampling would be warranted. Additional statistical methods were developed to demonstrate the use of short-term campaign measurements to characterize intermediate-scale (1 km) spatial variations of pollutant concentrations.

―――――――――――――――――――――

---

## Author Comment (AC3) · 24 Mar 2020

Interactive comment on "Mobile-Platform Measurement of Air Pollutant Concentrations in California: Performance Assessment, Statistical Methods for Evaluating Spatial Variations, and Spatial Representativeness by Paul A. Solomon et al.

Responses to Anonymous Referee #2

[Figure]

Thank-you for your constructive review. We summarize our responses to your questions and suggestions here. We will add these responses at appropriate places in the revised manuscript or in the supplement.

1. There is no discussion or comparison between Aclima instrumentation and capabilities to other sensors in the market (e.g. Purple Air), including technical and accuracy information. Have the authors done any comparison studies at similar times and locations to demonstrate Aclima outperforming other sensors?

Our study used only measurements from research-grade instruments (lines 78 – 79) but we have conducted sampling efforts using sensors during the past year. When we analyze our sensor data, we will attempt to compare them with other sensor-based studies. For this manuscript, we focused on comparisons to EPA-approved equipment at stationary sites in Los Angeles. We will add a table of technical specifications for our instruments. The instrumental methods, resolution, ranges, and response times are listed in the Lunden and LaFranchi (2017) citation and can be added to our supplement.

2. This analysis provides information on mobile air quality monitoring in a certain environment. The measurements represent air quality in urban locations near roads and that covers certain points/line measurements yet does not create a continuous air quality map.

Please see the proposed new paragraph in our "General Response" reply.

3. PN is measured by the Aclima platform for different size bins. It is not clear how this measurement is evaluated, as the EPA monitors particulate matter mass concentration?

We were not able to do an "apples-to-apples" field comparison to EPA monitors, as the reviewer notes. Nor could we do laboratory zero and span checks, as was done for the gas instrumentation. Instead, PN measurements were evaluated as described in lines 148 – 153.

[Figure]

4. It is not clear why the distance between cars is important in the discussion.

We will add text to discuss the utility and limitations of considering intervehicle variability versus distance. Please see also the proposed new paragraph in our "General Response" reply. Because our study was conducted as a series of short-term campaigns in several widely separated geographical areas, we did not attempt to develop pollutant maps that represent long-term concentration averages and which could be used to characterize spatial variations. Our study was conducted as a series of geographically separated sampling campaigns between May 2016 and September 2017, generally lacking the number of repeated driving routes needed to generate stable, long-term pollution maps. Instead, we used statistical metrics, such as FAMD, to characterize the spatial heterogeneity of pollutant concentrations. Because vehicles sample different road segments on different days and at different times of day, we compiled time-synchronized differences between the concentrations measured by two cars to remove the confounding effects of day-to-day and diurnal variability. Random differences between vehicles, such as short, intermittent exposures of one car or the other car to a high emitter, are averaged out in the FAMD statistic. In contrast, systematic car-to-car differences yield higher FAMD values. Systematic differences could occur if the instrumentation in one car was biased relative to the other car. After eliminating that source of systemic car-to-car difference through the side-by-side sampling comparisons, we can conclude that larger FAMD values (e.g., > 0.20 or 20%) represent spatial heterogeneity, e.g., due to the two cars sampling different neighborhoods (as indicated in Figure 6a or in Figures S6 and S10). Considering the relationship between FAMD and distance on a small (1 − 10) number of days provides a measure of the spatial scales over which concentrations changed by more than a specified amount (e.g., 20%). This is a useful metric for evaluating the spatial scale of representativeness of stationary monitors, for example. The relationship between FAMD and distance does not, of course, indicate which neighborhoods experienced higher pollutant concentrations. For that purpose, we developed the visualization shown in Figure 6.

5. All the measurements have been done for periods of several weeks and there is no 'long-term' monitoring campaign presented (e.g >1 year) that captures, for example, seasonality. This limitation of measurements period should be addressed in the discussion.

Please see our "General Response" and our response to #5. We will add both to our revised manuscript.

6. A description of the climatology at the different measurement locations is missing (e.g. temp, RH, and wind profiles, built area, type of road, no. of cars etc.). That can help understand some of the results.

By focusing on time-synchronous car-to-car measurement differences, we ensure that both vehicles are experiencing the same meteorological conditions. The figures and photos (Figures 6 and 7; Figures S3 – S4, S6 – S10) provide an indication of road density, built area, and proximity of driving routes to freeways. Population data for cities in the San Joaquin Valley are provided in lines 437 – 440 to complement Figure 7. Figures S11 – S15 indicate when the driving routes were in San Joaquin Valley cities and when they were on freeways. We will add text to better highlight how this information was used, or can be used, to help interpret the results.

7. The authors should do a better job in stating the limitations of the Aclima platform in this study set and in general.

Please see our "General Response."

8. Did the authors consider validating their results with continuous modeled data (CMAQ)? Or satellite data?

Because we focused on interpreting the results of a series of short-term campaigns, we did not compile pollution maps. Comparison of pollution maps generated from stable, long-term data to satellite data or modeling predictions could indeed provide complementary corroborating results. For such a comparison, one challenge would be

the incommensurability of the fine-scale mobile data and the coarser spatial scales of gridded modeling output or satellite imagery. The mobile data would need to be aggregated to the coarser scales for the comparison. Presumably, if results on consistent spatial scales were reasonably consistent, it would then be valuable to compare mobile monitoring maps generated from spatially-aggregated and -disaggregated data to better understand what is gained by the high-resolution mobile sampling.
* * *

---

## Author Comment (AC4) · 24 Mar 2020

Interactive comment on "Mobile-Platform Measurement of Air Pollutant Concentrations in California: Performance Assessment, Statistical Methods for Evaluating Spatial Variations, and Spatial Representativeness by Paul A. Solomon et al.

Responses to Anonymous Referee #3

[Figure]

Thank-you for your thorough review. We summarize our responses to your questions and suggestions here. We will add these responses at appropriate places in the revised manuscript or in the supplement. Please note the new paragraph that we propose to add to the introduction, which has been posted as "AC1 Authors' general response." We organize our responses by the line numbers noted in the review under the two review categories

Specific Comments:

Line 17: The time frame is misleading. Should indicate a few intensive (i.e. week to month long) campaigns were performed between May 2016 and September 2017.

Revise to: On-road measurements of air quality were made during a series of sampling campaigns between May 2016 and September 2017 at high. . .

Lines 20-21: Lifetime of NO2 is hours and O3 is >20 days in the troposphere. Observing the diurnal cycle and weekday weekend trends may be more appropriate than looking at a fortnight.

Our focus was on characterizing spatial rather than temporal variations. As noted in the proposed new paragraph, the cars drive weekdays between ∼9 a.m. and 5 p.m. The sampling regime therefore does not permit weekday/weekend comparisons nor does it lend itself to fully characterizing diurnal cycles.

Line 22: In-situ instrument or research-grade instruments. I'm sure they mean their instrument package.

Revise to: . . . research instruments located within stationary vehicles. . .

Line 31: Percentages of what? Concentration deltas?

Revise (here and at line 424) to: 1-km scale differences in NO2 and O3 concentrations up to 117% and 46%, respectively, of mean values

Line 75: Concentration decay rate from a point source will be highly variable and based

on several meteorological parameters.

Revise to: . . .could help establish concentration decay rates of mobile emissions with distance. . .

Line: 109-110: What are the limitations of overnight calibration when cars are parked next to each other?

We do not understand this question. Lines 109 – 110 simply state that we used the QD1 data because QD1 data sets included the time periods when the cars were parked next to each other, whereas such times had been filtered out of the QD2 data set. The instruments were switched from vehicle to line power when parked.

Lines 106,109/110, 113: The first lines seem to imply the mobile platform intercomparison was made overnight, 113 implies it may have been only a short period (5 min, 30 min), the SI from their Apte et al. indicates it was several hours overnight. Not sure about an intercomparison in a parking garage either, especially if it was during a time when vehicles were entering or leaving (cold starts vs operating temp emissions).

Revise to: . . . with additional sampling occurring while vehicles were parked in San Francisco and Los Angeles before ($\sim$ 6 – 9 a.m.) and after ($\sim$5 – 10 p.m.) driving periods.

The vehicles are parked away from traffic in the San Francisco parking garage. They were parked overnight in a small ($\sim$10 car) lot in Los Angeles.

Line 119: Was the audit the same as is done with FRM/FEM monitors via the National Performance Audit Program (NPAP)?

We will provide more detail to reduce possible confusion as to how the audits were accomplished.

Line 122: In table 4, there needs to be an explanation of the scales and how EPA initially established each sitting.

We will add citation to Appendix D to Part 58 - Network Design Criteria for Ambient Air Quality Monitoring (https://www.law.cornell.edu/cfr/text/40/appendix-D_to_part_58). The EPA scales are defined in Footnote 1 of Table 4.

Line 125: Suggested to add sentence or phrase to cover why the other stations were not used.

Revise lines 119 – 125 to: During the Los Angeles sampling, the South Coast Air Quality Monitoring District (SCAQMD) conducted calibration checks when the sampling vehicles were parked adjacent to stationary air quality monitoring sites (Table 3). The SCAQMD also prepared 1-minute resolution data files for measurements made at these and other stationary air quality monitoring sites (Table 4; see also location map, Figure S1). Data from one of the dates and locations (LAXH, September 20, 2016) were suitable for collocated comparison with mobile measurements (Table 3). The stationary-monitor data from W710 consisted only of 1-hour resolution PM2.5 mass (Table 4), which was not measured by the mobile platforms, and no data were provided for the Santa Clarita site (Tables 3).

Line 137: Do the monitoring stations use the Network Time Protocol? If not, address discrepancies this may cause in measurement comparisons.

We used time series plots, such as Figure S5, to confirm the alignment of mobile and station minima and maxima. The results indicate that any discrepancies are less than the 1-minute averaging times.

Line 164: What is the impact of wind and GPS location uncertainties on data collected while stationary?

When cars were parked adjacent to each other, we do not expect GPS location uncertainties or variations in wind sped or direction to impact the side-by-side comparisons.

Sec. 2.3: Are there multiple BC and CH4 instruments or just one that was moved between cars? I'm assuming this was done because of inlet restrictions.

One car was equipped with a BC instrument and one car was equipped with a CH4 instrument. There are three cars, though. The second BC and CH4 instruments were used by Apte et al. (2017) in their study. Since all vehicles parked in the same San Francisco garage, there were two BC and two CH4 instruments available for the side-by-side parked comparisons (line 180, Tables 5 and 6).

Line 173: Where are the reported variabilities of the paired differences shown?

These are reported in "Results and Discussion," rather than "Methods" (Section 3.2, Table 9).

Sec. 2.3: Was CARB contacted to ensure BC and CH4 observations were not present at sites? Some EPA sites have but don't advertise these observations.

For the field comparisons, we worked with South Coast Air Quality Management District staff, who operate the air quality monitors and are familiar with all measurements made.

Lines: 247- 249: Comparing different regions during different time periods without a detailed study of the meteorology is misleading if talking about local or neighborhood scales. Here are the climatological winds near San Joaquin Valley for March and November using data between 1973-2019 at Buchanan Field Airport in Concord, CA. Figure 1 shows the month of March may be experiencing inflow from the Chevron processing plant in Richmond and dust (Coarse mode, not reported) from Dutra Materials quarry in McNears Beach, while to a much lesser extent in November (Figure 2). Since the data is presented as mean concentrations during the sampling periods, I'd bet the baseline PN concentrations are different for the two months.

Lines 247 – 260 provide a summary overview of the measurements. These are useful but require caveats for various reasons such as those indicated by the referee. We stated at line 250, "Although differences in the PN distributions possibly reflect spatial variability, they more likely reflect seasonal variations in PM composition" and at lines 261 – 262, "As with PN, these average concentrations likely vary due to time of year,

location relative to source emissions, and chemical processing." For later analyses, we focused on time-synchronous differences between measurements made by two vehicles, not averages over short-term campaigns.

Lines 251-252: The deployed optical particle counter provided five size ranges why report only the smallest, then reference a paper regarding a measured size bin that was not reported in the paper?

Reasons for focusing on the smallest size fraction were explained in the previous paragraph, lines 240-246.

Lines 246-262: I'm not sure this section is representative and should be included here and should likely be absorbed by the following sections.

As noted previously, this paragraph provides a summary overview of the measurements but is not the basis for our analyses of spatial variability.

Line 265: Typo. . . . vehicles drove in the Los Angeles

Revise by removing "the"

Line 270: Why was the mean relative difference between the two calibrations so high? An absolute difference of 5% NPAP would require corrective actions. The calibration gases and flow meters used should be traceable to NIST for re-evaluation.

We do not see large (>5%) differences between the internal and external calibration checks in Table 8. The invalidating limits for the South Coast Air Quality Management District's weekly calibration checks are 7% for O3 and 10% for CO, SO2, and NOx, warning limits are 5% for O3 and 7% for CO, SO2, and NOx (Table 2.4, https://ww3.arb.ca.gov/aaqm/qa/pqao/repository/district_sops/south_coast/quality_assurance/qapp_criteria_pollutants.pdf

Sec. 3.2: When comparing inter-vehicle observations were the vehicles traveling the same route (i.e. following each other) or just driving the same neighborhoods and passing by each other?

The cars generally followed different routes, such as those illustrated in Figure 7 or in Figures S3, S6 – S8, and S10 – S15. When the cars traveled a route segment together (e.g., Figure S15), they traveled "caravan style", keeping each other in sight but not following immediately one behind the other.

Line 294: Were the vehicles were running during the LAXH comparison or were the instruments moved to shelter power and the vehicle engines shutoff?

The instruments were switched from vehicle to line power in the parking garage or parking lot but this option was not logistically practical for the one-day comparison at LAXH.

Sec. 3.3: Last sentence of section, CH4 emissions from vehicles is extremely small (something like <0.2% of anthropogenic emissions) and the lifetime of NO very short. This statement needs a citation, or it needs to be removed.

Add citation: Nam et al., ES&T, 2004 "We recommend the use of an average emission factor for the U.S. on-road vehicle fleet of (g of CH4/g of CO2) ) = (15± 4)x10-5 and estimate that the global vehicle fleet emits 0.45± 0.12 Tg of CH4 yr-1 (0.34± 0.09 Tg of C yr-1), which represents <0.2% of anthropogenic CH4 emissions." https://pubs.acs.org/doi/10.1021/es034837g. We agree that NO has a short residence time compared to CH4. However, a correlation between NO and CH4 will be observed when sampling fresh automotive exhaust emissions.

Sec 3.4: This section will have a very large dependence on meteorological parameters.

We agree that meteorology impacts the concentrations measured at two distant points, as do emission sources and chemical and physical processing. The last sentence of the first paragraph in section 3.4 states "The intent of the analyses in this section is to help elucidate the spatial scales over which stationary-monitor and mobile-platform data represent ambient concentrations." The analyses utilize time-synchronous differences so that each vehicle is experiencing the same meteorological conditions.

Sec. 3.4.1: The air masses the vehicles are sampling are potentially different. An intervehicle comparison could be made in time and latitude. As it is, the comparisons are meaningless because we know the location of any vehicle at any given time and one may be sampling south of the Santa Ana Freeway and the other sampling all three major N-S freeways in the area. The attached Figure 3 shows winds are between 9am and 5pm averaged over Aug 3-16. See comment above, section 3.4.

Because vehicles sample different road segments on different days and at different times of day, we compiled time-synchronous differences between the concentrations measured by two cars to remove the confounding effects of day-to-day and diurnal variability. Random differences between vehicles, such as short, intermittent exposures of one car or the other car to a high emitter or variations in wind directions, are averaged out in the FAMD statistic. In contrast, systematic car-to-car differences yield higher FAMD values. Systematic differences could occur if the instrumentation in one car was biased relative to the other car. After eliminating that source of systemic car-to-car difference through the side-by-side sampling comparisons, we can conclude that larger FAMD values (e.g., > 0.20 or 20%) represent spatial heterogeneity, e.g., due to the two cars sampling different neighborhoods (as indicated in Figure 6a or in Figures S6 and S10). FAMD is a useful metric for evaluating the spatial scale of representativeness of stationary monitors, for example. The relationship between FAMD and distance does not, of course, indicate which neighborhoods experienced higher pollutant concentrations. For that purpose, we examined maps, such as shown in Figures S6 and S10, and photos such as those provided by the referee; we also developed the visualization shown in Figure 6. Please note also our interpretation at lines 352 – 355.

Line 362: Driving near as in right past along Dowlen Dr or within n meters? Wilshire Blvd is ∼200m as is Federal Ave.

Line 360 defines "near" as 0.5 to 5 km. The routes include all areas shown in Figure S8. We will revise the text for clarity.

[Figure]

Line 392: What grid is used?

Nearest kilometer as calculated by conversion of latitude and longitude to UTM coordinates.

Fig. 6: Needs legend, different colors for positive and negative intervehicle differences and FMD differences not red/blue, which were used to identify specific vehicles in the same figure.

We will revise this figure.

Line 424: Enhancements based on what? FAMD is comparing observations at the same time, is the enhancement based on location as stated in the paragraph before or between May 1-12?

Revise to: 1-km scale differences in NO2 and O3 concentrations up to 117% and 46%, respectively, of mean values

Line 440: Routes for November 16th, 2016 are not in SI but referenced in text.

Revise to: The initial drives occurred November 16 – 23, 2016 (Figure 7, November 16; see also example of drives on other days in Figures S11 – S15).

Include Line 457: Are traffic count data available?

We did not use traffic count data in our analyses but they are available.

Line 459: Enhancements compared to what, background?

Lines 459 – 460 state "…enhancements of pollutant concentrations in northern San Joaquin Valley cities over concentrations occurring in surrounding areas

Line 529: Enhancements based on what?

Revise to: 1-km scale differences in NO2 and O3 concentrations up to 117% and 46%, respectively, of mean values

General: Overall distance bins should be the same for all missions. Seems like all the analysis times were weekday (do Google Street View vehicles drive on weekends)?

The spatial scales of the sampling routes differed among the missions, so the distance bins also differ. As noted at line 112, measurements were made between $\sim$ 9 a.m. and 5 p.m. on weekdays. We identify this as a limitation in the new paragraph posted as "AC1."

TECHNICAL CORRECTIONS:

Line 35: Suggested to add spatial variability context for pollutants to introduction as this has implications on reported uncertainties. Seems this is provided starting at about line 48 of the intro.

Lines 35 – 60 provide this context. It isn't evident that reordering sentences would improve clarity.

Line 155: LOD is defined in Table 5 subtext, but not in text. Consider defining in main text.

Revise line 153 to: We calculate BC limit of detection (LOD) (see footnote 2, Table 5) using data reported. . .

Lines 172-174: Suggested to remove 'merge' detail, as it seems superfluous to the reader, and combine the two sentences into one focusing on temporally coincident pairing.

Revise to: Data files were merged by 1-s or 1-minute resolution times and were then used to determine time-matched paired differences, which were evaluated as functions of ambient concentration, intervehicle distance, and vehicle speed.

Lines 185, 190, 195, 200: 'Car B Difference' could be misleading. It is suggested to move the word 'Difference' to after the word 'Mean' (i.e., Mean Difference) or use wording such as 'Mean [Absolute] Difference between Car A and B' in the numerator.

[Figure]

These changes will be made.

Line 206: Z is not defined.

Z is simply an example variable, not a measurement. Lines 205 – 209 will be replaced by simple citations.

Line 216: MD already defined in line 185.

Not redefining MD, just restating for clarity, revise to: MD is used . . .

Lines 211 and 222: Consistency in section references.

Capitalize in both locations.

---

## Author Comment (AC5) · 26 Mar 2020

Please note two corrections and one clarification to AC4. First, the overnight parking lot in Los Angeles holds ∼30 vehicles. Second, there are four Google vehicles equipped with the Aclima platform. Two vehicles were available for our study, with a third available during part of the San Joaquin sampling. Of the two vehicles used throughout our study, one was equipped with methane instrumentation and one with

a black carbon (BC) instrument. The other two vehicles were each equipped with BC instrumentation, thereby providing side-by-side BC comparisons while parked in the San Francisco garage (Table 6). Finally, measurements of all parameters were made and recorded while vehicles were parked overnight in San Francisco and Los Angeles between $\sim$ 6 - 9 a.m. before starting the drive routes and between $\sim$ 5 - 10 p.m. after returning.

---

## Author Response (AR1)

**Summary of Revisions**

Thank-you for your reviews, which have helped us improve the manuscript. We have incorporated responses to all review questions and suggestions in the revised manuscript.

In all three reviews, several questions occurred about two topics: measurement limitations and the statistical approach of differencing time-synchronized concentrations. We have addressed these questions at appropriate places in the manuscript. We also provided more context and explanation by expanding the brief overview description of the study and study objectives that appears at the end of the introduction (lines 78 - 100), where we added the following paragraph.

"The mobile sampling discussed here and in Apte et al. (2017) is limited to weekdays between ~9 a.m. and 5 p.m. Sampling is necessarily conducted along roads and streets. Depending on the number of repeated driving segments, vehicles sample different road segments on different days or at different times of day. These limitations are important considerations for studies whose goal is to develop pollutant maps that represent long-term concentration averages, and which are intended to correctly characterize spatial variations at a desired spatial scale. Our objectives are different, however. The principal objectives of our

- 15 study are to examine the capabilities of research instruments when placed in stationary and moving vehicles, to compare our measurements with those obtained from stationary air quality monitors, to evaluate driving and sampling strategies, and to develop statistical methods that account for sampling limitations. Limitations that are specific to our study are that (1) it was conducted as a series of geographically separated sampling campaigns between May 2016 and September 2017, generally lacking
- 20 the number of repeated driving routes previously used to generate pollution maps (Apte et al., 2017; Messier et al., 2018), and (2) no collection of driving routes completely covered any specific geographical domain (e.g., San Francisco or specific neighborhoods therein). The results presented here therefore focus on measurement and methodological questions that can be addressed with data available from the individual sampling campaigns. A set of research questions was developed initially and was then used to
- 25 design the individual sampling campaigns. In analyzing the results, a need arose to distinguish between temporal variability (due, e.g., to sampling different places at different times) and spatial variability. Statistical methods were therefore developed to characterize spatial heterogeneity within and between neighborhoods by utilizing time-synchronized differences in the pollutant concentrations that were measured by different vehicles. Due to limited repeated sampling of individual road segments, our
- 30 estimates of spatial heterogeneity do not in themselves identify specific spatial coordinates of long-term high and low pollutant concentrations. However, areas with high spatial heterogeneity indicate where more intense future sampling would be warranted. Additional statistical methods were developed to demonstrate the use of short-term campaign measurements to characterize intermediate-scale (1 km) spatial variations of pollutant concentrations."

**Responses to Anonymous Referee #1**

Thank-you for your helpful questions and suggestions. We summarize our responses here. We added these responses at appropriate places in the revised manuscript or in the supplement.

1. Some information on the cars should be included, not just referenced in another paper. Were the cars'

40 engines running while parked (e.g., in the garage, near stationary monitors, etc.)? What was used to power the instruments?

The instruments were switched from vehicle to line power in the parking garage or parking lot.

2. It is great that the inlets were designed to minimize self-sampling, but were additional steps taken during post-processing to remove potential periods of self sampling, or of sampling the Google car in front?

45 front?

50

65

The cars generally followed different routes, such as those illustrated in Figure 7 or in Figures S3, S6 - S8, and S10 - S15. Therefore, sampling the exhaust of a partner car was seldom an issue. When the cars traveled a route segment together (e.g., Figure S15), they could not travel side-by-side because that would block the flow of traffic. The drivers instead traveled "caravan style", keeping each other in sight but not following immediately one behind the other.

3. It would be good to document the limitations of the study (e.g., no overnight monitoring on roads or in early morning when the boundary layer is likely at its lowest).

Please see the new paragraph above and following line 77 in the revision.

4. Table 1 - was winter included in the San Joaquin Valley measurements (Nov '16-Apr '17) or was it just fall and spring?

Sampling was conducted in the San Joaquin Valley between November 2016 and April 2017. We did not attempt to analyze the full set of measurements because another manuscript will likely be needed to fully describe the intracity, intercity, and urban-rural differences encountered in this geographically large domain. No single area was sampled throughout the entire period.

5. Section 2.2 - were the cars parked on the roof of the parking structures or on a lower level? Depending, this could explain why GPS uncertainties were not comparable to manufacturer specs at times. Tall buildings nearby (if present) would also impact GPS performance.

In San Francisco, the cars were parked within a parking structure. In Los Angeles, the cars were parked in a small (~30 car) open parking lot. We corrected this statement (e.g., at old lines 109, 113, 165) in the revision. We do not have an explanation for the observed variations in the GPS location uncertainties,

but report them for completeness.

6. Lines 286-289: the closely-spaced moving vehicle condition makes it highly likely that the following car is measuring exhaust emissions from the lead car. This point should be mentioned in the manuscript. Did you try to correct for this? Why not drive side-by-side (road permitting)?

**70 Please see our response to #2.**

7. Could you please provide a list or table in the SI with the manufacturer and model of all instruments used in the study along with response time and measurement frequency. Even if this information was referenced in another paper, it should be reported here.

We added this information to the supplement (new Table S1). The instrumental methods, resolution, ranges, and response times are from the Lunden and LaFranchi (2017) citation.

8. Did you sync all instruments to the same time standard before measurements? Did you check instrument times at the completion of each day's measurement to quantify time drift? At measurement frequency of 1Hz, time drift can have a major impact on data comparison. These details should be included in the methods section.

- 80 We clarified this point at old line 137. The on-board computers are synchronized throughout the day using network time protocol, which synchronizes computers to Coordinated Universal Time (UTC) with accuracies on the order of milliseconds. This approach was necessary to ensure that the 1 Hz measurements did not drift in time.
- 9. By comparing 1-min averages, which I understand is important in order to maintain higher spatial resolution, how are you able to separate out the spatial trends due to differences in regional concentrations as opposed to differences in measurements due to some very localized conditions (e.g., driving behind a truck for a short period of time with one Google car but not the other over the same time period)? Would some other comparisons be more appropriate, such as a 60-second moving 5th percentile, or something comparable, to smooth out hyper-local concentrations?
- 90 Several different comparisons were made at 1-minute or coarser resolution. For the Los Angeles car-tocar comparisons, we examined both bin-average FAMD and variability within bin averages. Random differences between vehicles, such as short, intermittent exposures of one car or the other car to a high emitter, are averaged out in the FAMD statistic. In contrast, systematic car-to-car differences yield higher FAMD values. We identified some specific geographical patterns associated with higher FAMD (old lines
- 95 352 355). As noted, the approach developed for studying the San Francisco data could also be applied to the Los Angeles data for a more comprehensive analysis.

For the San Francisco data, we aggregated 1-minute differences to a 1-km spatial scale (old lines 402 - 410). Large mean differences were plotted in Figure 6 only if they were statistically different from zero (i.e., the interval of the mean difference  $\pm 2$  standard errors of the mean did not cover zero) so that

100 atypical car-to-car comparisons did not artificially create apparent spatial patterns. The rationale is that the standard errors of the 1-km averages would be large if one or more paired differences was very large; this would indicate the occurrence of an unusual condition.

10. I am struggling to understand why plotting the measurements against distance between either the cars or between car and stationary monitor is the best way to present the data. Had the cars been driving
 different routes than the ones presented, the plots would be completely different? The distance between

the cars is not driving the differences observed, it is the difference in the environments of the two cars at any given time. For example, the cars could both be in heavy traffic at 50 km away from each other (thus mean differences in concentrations are low), then at 75 km distance one car is still in heavy traffic while the other is in a quiet neighborhood away from highways (thus mean differences in concentrations are

- 110 high). For example, Figures S16 and S17 are interesting, but it would be more informative to provide information on where each of the cars are (e.g., land use, traffic conditions, major roadway, etc.) when FAMD is higher or lower irregardless of the distance between the cars. Are all points where the cars are X distance away from each other aggregated together even if the positions were discontinuous? If so, I do not know how one could interpret this plot.
- 115 We added text in the introduction and in preceding the definitions of the statistical metrics to explain how we can interpret the plots of differences versus intervehicle distance to characterize the spatial scales of pollutant heterogeneity. This issue clearly affects the data from the San Joaquin Valley, where the cars were separated by larger distances than in Los Angeles or San Francisco. As noted by the reviewer and as shown in Figures S15 and S16, for some species, differences in environments can drive car-to-car
- 120 differences when the vehicles are separated by more than a few kilometers. In contrast, Figure S17 shows the expected regional character of ozone with FAMD values < 0.2 at all intervehicle distances < 50 km. We conclude that smaller FAMD values indicate greater spatial homogeneity; larger FAMD values require further study beyond the plots of FAMD vs. distance. High FAMD values indicate where further study would be informative. As one example, Tables S2 – S5 summarize car-to-car comparisons that are
- 125 stratified by sampled areas. We expect that more complete analyses of the San Joaquin Valley data will be quite informative but will require another manuscript to fully explore.

11. Section 3.6 (lines 476-479): An FAMD of 0.5 seems high to conclude that a reference monitor is representative of a neighborhood scale area.

We revised this statement to note that the majority of the  $NO_2$  FAMD values were less than 0.2 at car-130 monitor distances of 0.5 - 4 km. We noted the higher FAMD for NO. EPA defines this monitor as neighborhood scale for  $O_3$  and  $NO_2$  but not NO.

**Responses to Anonymous Referee #2**

Thank-you for your constructive review. We summarize our responses to your questions and suggestions here. We have added these responses at appropriate places in the revised manuscript or the supplement.

1. There is no discussion or comparison between Aclima instrumentation and capabilities to other sensors in the market (e.g. Purple Air), including technical and accuracy information. Have the authors done any comparison studies at similar times and locations to demonstrate Aclima outperforming other sensors?

Our study used only measurements from research-grade instruments (lines 78 – 79) and we added instrument specifications as new Table S1. Aclima has conducted sampling efforts using sensors during the past year. When the sensor data are analyzed, they can be compared to other sensor-based studies. For this manuscript, we focused on comparisons to EPA-approved equipment at stationary sites in Los Angeles.

2. This analysis provides information on mobile air quality monitoring in a certain environment. The
 measurements represent air quality in urban locations near roads and that covers certain points/line
 measurements yet does not create a continuous air quality map.

Please see the new paragraph above and in the introduction at old line 77.

3. PN is measured by the Aclima platform for different size bins. It is not clear how this measurement is evaluated, as the EPA monitors particulate matter mass concentration?

150 PN measurements were evaluated as described in old lines 148 – 153. We were not able to do an "applesto-apples" field comparison to EPA monitors, as the reviewer notes. Nor could we do laboratory zero and span checks, as was done for the gas instrumentation.

4. It is not clear why the distance between cars is important in the discussion.

- We added text to discuss the utility and limitations of intervehicle variability versus distance. Please see also the new paragraph in the introduction (line 79). Because our study was conducted as a series of short-term campaigns in several widely separated geographical areas, we did not attempt to develop pollutant maps that represent long-term concentration averages and which could be used to characterize spatial variations. Our study was conducted as a series of geographically separated sampling campaigns between May 2016 and September 2017, generally lacking the number of repeated driving routes needed
- 160 to generate stable, long-term pollution maps. Instead, we used statistical metrics, such as FAMD, to characterize the spatial heterogeneity of pollutant concentrations. Because vehicles sample different road segments on different days and at different times of day, we compiled time-synchronized differences between the concentrations measured by two cars to remove the confounding effects of day-to-day and diurnal variability. Random differences between vehicles, such as short, intermittent exposures of one car
- 165 or the other car to a high emitter, are averaged out in the FAMD statistic. In contrast, systematic car-tocar differences yield higher FAMD values. Systematic differences could occur if the instrumentation in one car was biased relative to the other car. After eliminating that source of systemic car-to-car difference through the side-by-side sampling comparisons, we can conclude that larger FAMD values (e.g., > 0.20 or 20%) represent spatial heterogeneity, e.g., due to the two cars sampling different neighborhoods (as
- 170 indicated in Figure 6a or in Figures S6 and S10). Considering the relationship between FAMD and distance on a small (1 - 10) number of days provides a measure of the spatial scales over which concentrations changed by more than a specified amount (e.g., 20%). This is a useful metric for evaluating the spatial scale of representativeness of stationary monitors, for example. The relationship between FAMD and distance does not, of course, indicate which neighborhoods experienced higher
- 175 pollutant concentrations. For that purpose, we developed the visualization shown in Figure 6.

5. All the measurements have been done for periods of several weeks and there is no 'long-term' monitoring campaign presented (e.g >1 year) that captures, for example, seasonality. This limitation of measurements period should be addressed in the discussion.

Please see the new paragraph in the introduction (line 79) and our response to #5.

180 6. A description of the climatology at the different measurement locations is missing (e.g. temp, RH, and wind profiles, built area, type of road, no. of cars etc.). That can help understand some of the results.

By focusing on time-synchronous car-to-car measurement differences, we ensure that both vehicles are experiencing the same meteorological conditions. The figures and photos (Figures 6 and 7; Figures S3 – S4, S6 – S10) provide an indication of road density, built area, and proximity of driving routes to freeways. Population data for cities in the San Joaquin Valley are provided in lines 437 - 440 to

complement Figure 7. Figures S11 – S15 indicate when the driving routes were in San Joaquin Valley cities and when they were on freeways. We added text to better highlight how this information was used, or can be used, to help interpret the results.

7. The authors should do a better job in stating the limitations of the Aclima platform in this study set and in general.

Please see the new paragraph in the introduction (line 79).

8. Did the authors consider validating their results with continuous modeled data (CMAQ)? Or satellite data?

- Because we focused on interpreting the results of a series of short-term campaigns, we did not compile pollution maps. Comparison of pollution maps generated from stable, long-term data to satellite data or modeling predictions could indeed provide complementary corroborating results. For such a comparison, one challenge would be the incommensurability of the fine-scale mobile data and the coarser spatial scales of gridded modeling output or satellite imagery. The mobile data would need to be aggregated to the coarser scales for the comparison. Presumably, if results on consistent spatial scales were reasonably
- 200 consistent, it would then be valuable to compare mobile monitoring maps generated from spatiallyaggregated and -disaggregated data to better understand what is gained by the high-resolution mobile sampling.

**Responses to Anonymous Referee #3**

185

205 Thank-you for your thorough review. We summarize our responses to your questions and suggestions here. We added these responses at appropriate places in the revised manuscript or in the supplement. Please note the new paragraph that we propose to add to the introduction.

We organize our responses by the line numbers noted in the review under the two review categories

**210 Specific Comments: -**

220

225

Line 17: The time frame is misleading. Should indicate a few intensive (i.e. week to month long) campaigns were performed between May 2016 and September 2017.

Revised to: On-road measurements of air quality were made during a series of sampling campaigns between May 2016 and September 2017 at high...

Lines 20-21: Lifetime of NO2 is hours and O3 is >20 days in the troposphere. Observing the diurnal cycle and weekday weekend trends may be more appropriate than looking at a fortnight.

Our focus was on characterizing spatial rather than temporal variations. As noted in the proposed new paragraph, the cars drive weekdays between ~9 a.m. and 5 p.m. The sampling regime therefore does not permit weekday/weekend comparisons nor does it lend itself to fully characterizing diurnal cycles.

Line 22: In-situ instrument or research-grade instruments. I'm sure they mean their instrument package.

Revised to: ... research instruments located within stationary vehicles...

Line 31: Percentages of what? Concentration deltas?

*Revised (here and at line 424) to: 1-km scale differences in*  $NO_2$  *and*  $O_3$  *concentrations up to 117% and 46%, respectively, of mean values*

Line 75: Concentration decay rate from a point source will be highly variable and based on several meteorological parameters.

Revised to: ...could help establish concentration decay rates of mobile emissions with distance...

Line: 109-110: What are the limitations of overnight calibration when cars are parked next to each other?

230 Lines 109 – 110 simply state that we used the QD1 data because QD1 data sets included the time periods when the cars were parked next to each other, whereas such times had been filtered out of the QD2 data set. The instruments were switched from vehicle to line power when parked.

Lines 106,109/110, 113: The first lines seem to imply the mobile platform intercomparison was made overnight, 113 implies it may have been only a short period (5 min, 30 min), the SI from their Apte et al.

235 indicates it was several hours overnight. Not sure about an intercomparison in a parking garage either, especially if it was during a time when vehicles were entering or leaving (cold starts vs operating temp emissions).

*Revised to:* ... with additional sampling occurring while vehicles were parked in San Francisco and Los Angeles before ( $\sim 6 - 9 \text{ a.m.}$ ) and after ( $\sim 5 - 10 \text{ p.m.}$ ) driving periods.

240 The vehicles are parked away from traffic in a designated area in the San Francisco parking garage. They were parked overnight in a small (~30 car) lot in Los Angeles. Line 119: Was the audit the same as is done with FRM/FEM monitors via the National Performance Audit Program (NPAP)?

245 Please see Section 3.1.

Line 122: In table 4, there needs to be an explanation of the scales and how EPA initially established each sitting.

We added citation to Appendix D to Part 58 - Network Design Criteria for Ambient Air Quality Monitoring (https://www.law.cornell.edu/cfr/text/40/appendix-D\_to\_part\_58). The EPA scales are 250 defined in Footnote 1 of Table 4.

Line 125: Suggested to add sentence or phrase to cover why the other stations were not used.

Revised lines 119 – 125 to: During the Los Angeles sampling, the South Coast Air Quality Monitoring District (SCAQMD) conducted calibration checks when the sampling vehicles were parked adjacent to stationary air quality monitoring sites (Table 3). The SCAQMD also prepared 1-minute resolution data

- 255 files for measurements made at these and other stationary air quality monitoring sites (Table 4; see also location map, Figure S1). Data from one of the dates and locations (LAXH, September 20, 2016) were suitable for collocated comparison with mobile measurements (Table 3). The stationary-monitor data from W710 consisted only of 1-hour resolution PM2.5 mass (Table 4), which was not measured by the mobile platforms, and no data were provided for the Santa Clarita site (Tables 3).
- 260 Line 137: Do the monitoring stations use the Network Time Protocol? If not, address discrepancies this may cause in measurement comparisons.

We used time series plots, such as Figure S5, to confirm the alignment of mobile and station minima and maxima. The results indicate that any temporal discrepancies are less than the 1-minute averaging times.

Line 164: What is the impact of wind and GPS location uncertainties on data collected while stationary?

265 When cars were parked adjacent to each other, we do not expect GPS location uncertainties or variations in wind speed or direction to impact the side-by-side comparisons.

Sec. 2.3: Are there multiple BC and CH4 instruments or just one that was moved between cars? I'm assuming this was done because of inlet restrictions.

One car was equipped with a BC instrument and one car was equipped with a CH4 instrument. There are four cars, though. Two cars were used by Apte et al. (2017) in their study. Since all vehicles parked in the same San Francisco garage, there were two BC and two CH4 instruments available for the side-byside parked comparisons (line 180, Tables 5 and 6).

Line 173: Where are the reported variabilities of the paired differences shown?

These are reported in "Results and Discussion," rather than "Methods" (Section 3.2, Table 9).

Sec. 2.3: Was CARB contacted to ensure BC and CH4 observations were not present at sites? Some EPA sites have but don't advertise these observations.

For the field comparisons, we worked with South Coast Air Quality Management District staff, who operate the air quality monitors and are familiar with all measurements made.

280 Lines: 247- 249: Comparing different regions during different time periods without a detailed study of the meteorology is misleading if talking about local or neighborhood scales.

Here are the climatological winds near San Joaquin Valley for March and November using data between 1973-2019 at Buchanan Field Airport in Concord, CA. Figure 1 shows the month of March may be experiencing inflow from the Chevron processing plant in Richmond and dust (Coarse mode, not reported) from Dutra Materials quarry in McNears Beach, while to a much lesser extent in November

285 reported) from Dutra Materials quarry in McNears Beach, while to a much lesser extent in November (Figure 2). Since the data is presented as mean concentrations during the sampling periods, I'd bet the baseline PN concentrations are different for the two months.

Lines 247 – 260 provide a summary overview of the measurements. These are useful but require caveats for various reasons such as those indicated by the referee. As we stated at line 250, "Although differences

- 290 in the PN distributions possibly reflect spatial variability, they more likely reflect seasonal variations in PM composition" and at lines 261 – 262, "As with PN, these average concentrations likely vary due to time of year, location relative to source emissions, and chemical processing." For clarity, we revised these lines to: "Although these differences in the PN size distributions possibly reflect regional-scale spatial variability, no simple comparison among regions is possible due to sampling them during different
- 295 seasons. The regional differences could reflect seasonal variations in PM composition: the observed variations in PN distributions are consistent with past studies that indicate the importance of PM nitrate (NO3) found in larger (> 0.5 μm) size fractions primarily as ammonium nitrate in California during cooler months (e.g., Herner et al., 2005), which could lead the observance of different size distributions in the different regions". For later analyses, we focused on time-synchronous differences between
- 300 measurements made by two vehicles, not averages over short-term campaigns.

Lines 251-252: The deployed optical particle counter provided five size ranges why report only the smallest, then reference a paper regarding a measured size bin that was not reported in the paper?

Please see previous response.

Lines 246-262: I'm not sure this section is representative and should be included here and should likely 305 be absorbed by the following sections.

As noted previously, this paragraph provides a summary overview of the measurements but is not the basis for our analyses of spatial variability.

Line 265: Typo. ... vehicles drove in the Los Angeles

310 Revised by removing "the"

335

Line 270: Why was the mean relative difference between the two calibrations so high? An absolute difference of 5% NPAP would require corrective actions. The calibration gases and flow meters used should be traceable to NIST for re-evaluation.

We do not see large (>5%) differences between the internal and external calibration checks in Table 8.
The invalidating limits for the South Coast Air Quality Management District's weekly calibration checks are 7% for O3 and 10% for CO, SO2, and NOx, warning limits are 5% for O3 and 7% for CO, SO2, and NOx (Table 2.4, https://ww3.arb.ca.gov/aaqm/qa/pqao/repository/district\_sops/south\_coast/quality\_assurance/qapp\_cri teria pollutants.pdf).

320 Sec. 3.2: When comparing inter-vehicle observations were the vehicles traveling the same route (i.e. following each other) or just driving the same neighborhoods and passing by each other?

The cars generally followed different routes, such as those illustrated in Figure 7 or in Figures S3, S6 – S8, and S10 – S15. When the cars traveled a route segment together (e.g., Figure S15), they traveled "caravan style", keeping each other in sight but not following immediately one behind the other.

325 Line 294: Were the vehicles were running during the LAXH comparison or were the instruments moved to shelter power and the vehicle engines shutoff?

The instruments were switched from vehicle to line power in the parking garage or parking lot but this option was not logistically practical for the one-day comparison at LAXH.

Sec. 3.3: Last sentence of section, CH4 emissions from vehicles is extremely small (something like <0.2%</li>
of anthropogenic emissions) and the lifetime of NO very short. This statement needs a citation, or it needs to be removed.

We added citation: Nam et al., ES&T, 2004 "We recommend the use of an average emission factor for the U.S. on-road vehicle fleet of (g of CH4/g of CO2) ) =  $(15\pm4)x10^{-5}$  and estimate that the global vehicle fleet emits  $0.45\pm0.12$  Tg of CH4 yr-1 ( $0.34\pm0.09$  Tg of C yr-1), which represents <0.2% of anthropogenic CH4 emissions." https://pubs.acs.org/doi/10.1021/es034837g.

We agree that NO has a short residence time compared to CH4. However, a correlation between NO and CH4 will be observed when sampling fresh automotive exhaust emissions. The revised sentence reads: "CH4 is reported in motor-vehicle emissions (Nam et a., 2004), so a correlation between NO and CH4 will usually be observed when sampling fresh automotive exhaust emissions; all NO values correlated with Coltrane CH4 concentrations ( $r^2 = 0.84$  to 0.87; Flora did not report CH4)."

Sec 3.4: This section will have a very large dependence on meteorological parameters.

We agree that meteorology impacts the concentrations measured at two distant points, as do emission sources and chemical and physical processing. The last sentence of the first paragraph in section 3.4 states "The intent of the analyses in this section is to help elucidate the spatial scales over which stationary-monitor and mobile-platform data represent ambient concentrations." The analyses utilize

345 stationary-monitor and mobile-platform data represent ambient concentrations." The analyses utilize time-synchronous differences so that each vehicle is experiencing the same meteorological conditions.

Sec. 3.4.1: The air masses the vehicles are sampling are potentially different. An intervehicle comparison could be made in time and latitude. As it is, the comparisons are meaningless because we know the location of any vehicle at any given time and one may be sampling south of the Santa Ana Freeway and the other sampling all three major N-S freeways in the area. The attached Figure 3 shows winds are between 9am and 5pm averaged over Aug 3-16. See comment above, section 3.4.

Because vehicles sample different road segments on different days and at different times of day, we compiled time-synchronous differences between the concentrations measured by two cars to remove the confounding effects of day-to-day and diurnal variability. Random differences between vehicles, such as

- 355 short, intermittent exposures of one car or the other car to a high emitter or variations in wind directions, are averaged out in the FAMD statistic. In contrast, systematic car-to-car differences yield higher FAMD values. Systematic differences could occur if the instrumentation in one car was biased relative to the other car. After eliminating that source of systemic car-to-car difference through the side-by-side sampling comparisons, we can conclude that larger FAMD values (e.g., > 0.20 or 20%) represent spatial
- 360 heterogeneity, e.g., due to the two cars sampling different neighborhoods (as indicated in Figure 6a or in Figures S6 and S10). FAMD is a useful metric for evaluating the spatial scale of representativeness of stationary monitors, for example. The relationship between FAMD and distance does not, of course, indicate which neighborhoods experienced higher pollutant concentrations. For that purpose, we examined maps, such as shown in Figures S6 and S10, and photos such as those provided by the referee;
- 365 we also developed the visualization shown in Figure 6. Please note also our interpretation at lines 352 355.

Line 362: Driving near as in right past along Dowlen Dr or within n meters? Wilshire Blvd is ~200m as is Federal Ave.

Revised to: Driving routes were near (<0.2 to 5 km) the west Los Angeles stationary monitor (WSLA,</li>
Table 4) on four of the 14 days between September 12 and 30 (including areas shown in Figure S8 for September 13 and 19; similar routes were driven on September 26 and 29).

Line 392: What grid is used?

Nearest kilometer as calculated by conversion of latitude and longitude to UTM coordinates. Revised to: "One-minute averages were next averaged spatially to the nearest kilometer (based on conversion of

375 latitude and longitude to Universal Transverse Mercator [UTM] coordinates) separately for each car (Figure 6b)..."

Fig. 6: Needs legend, different colors for positive and negative intervehicle differences and FMD differences not red/blue, which were used to identify specific vehicles in the same figure.

380 *Figure revised using different colors. Subpanels are defined in the caption.*

Line 424: Enhancements based on what? FAMD is comparing observations at the same time, is the enhancement based on location as stated in the paragraph before or between May 1-12?

*Revised to:* 1-km scale differences in  $NO_2$  and  $O_3$  concentrations up to 117% and 46%, respectively, of mean values

Line 440: Routes for November 16th, 2016 are not in SI but referenced in text.

Revised to: The initial drives occurred November 16-23, 2016 (Figure 7, November 16; see also example of drives on other days in Figures S11-S15).

Include Line 457: Are traffic count data available?

We did not use traffic count data in our analyses but they are available. Revised to: "High traffic volumes (~50,000 – 150,000 vehicles per day, annual average peak volumes) are typical of Highway 99

(https://dot.ca.gov/programs/traffic-operations/census/traffic-volumes, last access April 15, 2020), ... "

Line 459: Enhancements compared to what, background?

Lines 459 – 460 state "...enhancements of pollutant concentrations in northern San Joaquin Valley cities over concentrations occurring in surrounding areas"

395 Line 529: Enhancements based on what?

390

*Revised to:* 1-km scale differences in  $NO_2$  and  $O_3$  concentrations up to 117% and 46%, respectively, of mean values

General: Overall distance bins should be the same for all missions. Seems like all the analysis times were weekday (do Google Street View vehicles drive on weekends)?

400 The spatial scales of the sampling routes differed among the missions, so the distance bins also differ. As noted at line 112, measurements were made between ~ 9 a.m. and 5 p.m. on weekdays. We identify this as a limitation in the new paragraph in the introduction."

TECHNICAL CORRECTIONS: \_\_\_\_\_

Line 35: Suggested to add spatial variability context for pollutants to introduction as this has implications on reported uncertainties. Seems this is provided starting at about line 48 of the intro.

*Lines* 35 – 60 provide this context. It isn't evident that reordering sentences would improve clarity.

Line 155: LOD is defined in Table 5 subtext, but not in text. Consider defining in main text.

*Revised old line 153 to: We calculate BC limit of detection (LOD) (see footnote 2, Table 5) using data reported...*

410 Lines 172-174: Suggested to remove 'merge' detail, as it seems superfluous to the reader, and combine the two sentences into one focusing on temporally coincident pairing.

Revised to: Data files were merged by 1-s or 1-minute resolution times and were then used to determine time-matched paired differences, which were evaluated as functions of ambient concentration, intervehicle distance, and vehicle speed.

415 Lines 185, 190, 195, 200: 'Car B Difference' could be misleading. It is suggested to move the word 'Difference' to after the word 'Mean' (i.e., Mean Difference) or use wording such as 'Mean [Absolute] Difference between Car A and B' in the numerator.

These changes were made.

Line 206: Z is not defined.

420 *Z* is simply an example variable, not a measurement. Lines 205 – 209 were replaced with new citations and "Z" no longer is used.

Line 216: MD already defined in line 185.

Not meant to redefine MD, just restating for clarity; revised to remove "(MD)"

Lines 211 and 222: Consistency in section references.

425 Now capitalized in both locations.

[revised manuscript text omitted]

660 The variances σ2FMD, σ2FAMD, and σ2FMAD are derived from standard statistical formulae for propagating errors (e.g., variance of (X × Y) = {(σX × σY)2 + (σX × μY)2 + (σY × μX)2}, http://www.odelama.com/data analysis/Commonly Used Math Formulas/, last access September 27, 2019; Caldwell and Vahidsafa, 2019; Goodman, 1960; Ku, 1966), by transforming variables (X/Z = X × Y, Y = Z-1) and by making two assumptions: (1) the numerator and denominator (e.g., μA-B and μAB) are independent (implying zero covariance between differences and means), and (2) higher order terms (σ2X × σ2Y.) are small compared with

[revised manuscript text omitted]

  Q. i.k., EPA. 452/D. 18 002, https://university.com/parameter/parameter/parameter/parameter/parameter/parameter/parameter/parameter/parameter/parameter/parameter/parameter/parameter/parameter/parameter/parameter/parameter/parameter/parameter/parameter/parameter/parameter/parameter/parameter/parameter/parameter/parameter/parameter/parameter/parameter/parameter/parameter/parameter/parameter/parameter/parameter/parameter/parameter/parameter/parameter/parameter/parameter/parameter/parameter/parameter/parameter/parameter/parameter/parameter/parameter/parameter/parameter/parameter/parameter/parameter/parameter/parameter/parameter/parameter/parameter/parameter/parameter/parameter/parameter/parameter/parameter/parameter/parameter/parameter/parameter/parameter/parameter/parameter/parameter/parameter/parameter/parameter/parameter/parameter/parameter/parameter/parameter/parameter/parameter/parameter/parameter/parameter/parameter/parameter/parameter/parameter/parameter/parameter/parameter/parameter/parameter/parameter/parameter/parameter/parameter/parameter/parameter/parameter/parameter/parameter/parameter/parameter/parameter/parameter/parameter/parameter/parameter/parameter/parameter/parameter/parameter/parameter/parameter/parameter/parameter/parameter/parameter/parameter/parameter/parameter/parameter/parameter/parameter/parameter/parameter/parameter/parameter/parameter/parameter/parameter/parameter/parameter/parameter/parameter/parameter/parameter/parameter/parameter/parameter/parameter/parameter/parameter/parameter/parameter/parameter/parameter/parameter/parameter/parameter/parameter/parameter/parameter/parameter/parameter/parameter/parameter/parameter/parameter/parameter/parameter/para
  - Oxides, EPA-452/R-18-003, https://www.epa.gov/sites/production/files/2018-05/documents/primary\_so2\_naaqs\_-\_final\_rea\_-\_may\_2018.pdf (last access, June 24, 2019), 2018.

[revised manuscript text omitted]

2Neighborhood scale for O3; middle scale for other species

1230  $^{3}$  Middle scale for NO2; neighborhood scale for O3

4 Hourly PM2.5 or PM10 measurements available

| Pollutant (Car)            | Bias (ppbv) 1 | Precision (ppbv) 1 | Limit of Detection 2 (2σ, 1
sec) (ppbv) |
|----------------------------|--------------------------|-------------------------------|-------------------------------------------------------|
| NO (Coltrane)              | $\pm 2.1\% + 0.3$        | $\pm 2.3\% \pm 0.3$           | 1.5                                                   |
| NO (Flora)                 | $\pm 3.6\% + 0.3$        | $\pm4.3\%\pm0.3$              | 1.7                                                   |
| NO 2 (Coltrane) | $\pm~2.1\%~\pm~0.4$      | $\pm~2.8\%~\pm~0.5$           | <0.1                                                  |
| NO 2 (Flora)    | -2.4% + 0.2              | $\pm~2.2\%~\pm~0.2$           | <0.1                                                  |
| O 3 (Coltrane)  | $\pm~2.1\%~\pm~0.5$      | $\pm~2.4\%~\pm~0.6$           | 1.8                                                   |
| O 3 (Flora)     | $\pm~2.0\%~\pm~0.4$      | $\pm~2.3\%~\pm~0.5$           | 1.8                                                   |
| CH 4 (Coltrane) | ± 3.3                    | ± 3.3                         | n/a                                                   |

Table 5. Performance summary of the gas-phase instruments (NO, NO2, O3, and CH4) in parked vehicles (Lunden and LaFranchi, 2017).

[revised manuscript text omitted]

1 Sample sizes are total number of 1-sec measurements summed across vehicles. Means are weighted by the number of measurements per vehicle.

1270 2 Particle number in size fractions  $0.3 - 0.5 \mu m$ ,  $0.5 - 0.7 \mu m$ ,  $0.7 - 1.0 \mu m$ ,  $1.0 - 1.5 \mu m$ ,  $1.5 - 2.5 \mu m$ ,  $> 2.5 \mu m$ .

3 LA1 = Los Angeles, August 3 - 12, 2016 (8 days). BC, CH4 = 1 car; NO, O3, NO2, and PN = 2 cars.

4 LA2 = Los Angeles, September 12 – 30, 2016 (14 days). BC,  $CH_4 = 1$  car; NO, O3, NO2, and PN = 2 cars.

5 SJV1 = San Joaquin Valley, November 16 – 23, 2016 (6 days). BC,  $CH_4 = 1$  car, NO and  $O_3 = 2$  cars,  $NO_2$  and PN = 3 cars.

 $^{6}$  SJV2 = San Joaquin Valley, March 20 – 29, 2017 (6 days). BC, CH4 = 1 car, NO and O3 = 2 cars, NO2 and PN = 2 cars.

1275 7 SF = San Francisco, May 1 – 12, 2017 (10 days). BC,  $CH_4 = 1$  car; NO, O3, NO2, and PN = 2 cars.

Table 8. External calibration checks (zero and span) performed in Los Angeles with equipment and gas standards managed by the SCAQMD compared with internal checks performed by Aclima one month prior to the Los Angeles deployment, one month following this deployment, and during a 1-week return to San Francisco in the middle of the deployment. External and Aclima calibration checks were conducted through the inlet lines of the mobile platforms.

1280

| Species | Audit  | Bias                 | Precision           | Number of Span | Number of Zero |
|---------|--------|----------------------|---------------------|----------------|----------------|
|         |        | $(\% \pm ppbv)$      | $(\% \pm ppbv)$     | Checks         | Checks         |
| NO      | Aclima | $\pm 3.5\% + (< 1)$  | ± 4.5% + (< 1)      | 22             | 22             |
|         | SCAQMD | $\pm 8.2\% + (< 1)$  | $\pm 6.0\% + (< 1)$ | 10             | 10             |
| $NO_2$  | Aclima | $-3.7\% \pm 0.4^{1}$ | $\pm 3.7\% \pm 0.4$ | 19             | 20             |
|         | SCAQMD | - 1.9% $\pm 0.6^1$   | ± 4.9% ± 0.6        | 6              | 10             |
| $O_3$   | Aclima | ± 2.4% ± 0.9         | ± 2.3% ± 1.1        | 20             | 18             |
|         | SCAQMD | ± 3.3% ± 1.2         | ± 3.8% ± 1.5        | 10             | 10             |

1 Negative bias only

Table 9. Performance summary for measurements reported by collocated vehicles (mean difference ± 1 standard deviation; mean concentrations in parentheses). Standard deviations are reported here to indicate the variability of the 1-s differences. Mean differences provide a measure of average intervehicle differences. For periods when three vehicles were driven, the largest mean difference between vehicles is listed. The signs of the mean differences are not indicated because no vehicle is an audit standard. All values were determined from 1-s time resolution data.

[revised manuscript text omitted]

 $^{1}$  C = Coltrane, F = Flora, R = Rhodes